# Correlated Stochastic Block Models: Exact Graph Matching with Applications to Recovering Communities

**Miklós Z. Rácz**
Department of ORFE
Princeton University
Princeton, NJ 08544
mracz@princeton.edu

**Anirudh Sridhar**
Department of Electrical and Computer Engineering
Princeton University
Princeton, NJ 08544
anirudhs@princeton.edu

## Abstract

We consider the task of learning latent community structure from multiple correlated networks. First, we study the problem of learning the latent vertex correspondence between two edge-correlated stochastic block models, focusing on the regime where the average degree is logarithmic in the number of vertices. We derive the precise information-theoretic threshold for exact recovery: above the threshold there exists an estimator that outputs the true correspondence with probability close to 1, while below it no estimator can recover the true correspondence with probability bounded away from 0. As an application of our results, we show how one can exactly recover the latent communities using *multiple* correlated graphs in parameter regimes where it is information-theoretically impossible to do so using just a single graph.

## 1 Introduction

Learning community structure in networks is a ubiquitous inference task in several domains, including biology [13, 41], sociology [26], and machine learning [57, 37, 60]. Recent decades have therefore seen an explosion of work on the topic, leading to determining the fundamental information-theoretic limits for learning communities in probabilistic generative models [1, 2, 3, 45, 44, 42, 46], as well as algorithms that work well in practice [55, 33, 22]. Typically, such algorithms only leverage the structure of the network (i.e., the configuration of node-node links). Increasingly, one often has access to *side information* that can greatly improve the performance of inference algorithms.

There is a vast literature on designing algorithms that incorporate various types of side information to aid in recovering communities in networks. The works [21, 38, 47, 32, 8, 63, 11, 56, 39] leverage *node-level* information that is correlated with community memberships; here the sharp limits for community detection were conjectured by Deshpande et al. [21] and recently proven by Lu and Sen [38]. Another line of work [30, 5, 51, 52, 36, 4, 7, 43, 39] recovers communities from a *multi-layer* network, where the different layers are conditionally independent given the same community structure. Recently Ma and Nandy [39] synthesized these two strands of literature.

In contrast to prior work, we explore scenarios where the side information comes in the form of *multiple correlated networks*, which is natural in several domains including social networks [49, 53, 35], computational biology [59], and machine learning [15, 14]. In the context of social networks, for instance, many datasets are anonymized to protect the identity of users. Nevertheless, one may be able to infer additional information about users from additional networks by noting that the interaction patterns of the same set of users are likely to be similar across networks [49, 53, 35]. In computational biology, an important goal is to study the functional properties of protein groups

through a protein-protein interaction (PPI) network. Using the insight that functionally similar protein groups will have similar interaction structures, one can compare PPIs across species to infer protein functions [59]. In all of these examples, an important task, commonly known as *graph matching*, is to synthesize the information from multiple correlated networks in a sensible manner.

To the best of our knowledge, we are the first to consider the use of multiple correlated networks for recovering communities. Specifically, we quantify, in an information-theoretic sense, how much information we can gain from correlated networks in order to infer community structure. To this end, we focus on correlated graphs $G_1$ and $G_2$ drawn marginally according to the stochastic block model (SBM), which is widely recognized as the canonical probabilistic generative model for networks with community structure [31, 1]. The reason for studying this probabilistic model is twofold. For one, it serves as a prototypical model for networks with community structure found in practice, hence the algorithms we develop will serve as a starting point for applications. Moreover, the SBM has well-defined ground-truth communities, so we can concretely study the correctness of algorithms in terms of whether the communities they output align with the ground truth.

## 2 Models and Questions

**The stochastic block model (SBM).** The SBM is perhaps the simplest and most well-known probabilistic generative model for networks with community structure. It was initially proposed by Holland, Laskey, and Leinhardt [31] and subsequently used as a theoretical testbed for evaluating clustering algorithms on average-case networks (see, e.g., [24, 12, 9]). A striking fact about the SBM is that it exhibits sharp information-theoretic phase transitions for various inference tasks, leading to a *precise* understanding of when community information can be extracted from network data. Such phase transitions were first conjectured by Decelle et al. [20] and were subsequently proven rigorously by several authors [44, 42, 46, 2, 45, 3, 10, 1]. In summary, the SBM is a well-motivated and mathematically rich setting for studying inference tasks.

In this work we focus on the simplest setting, a SBM with two symmetric communities. For a positive integer $n$ and $p, q \in [0, 1]$, we construct $G \sim \mathrm{SBM}(n, p, q)$ as follows. The graph $G$ has $n$ vertices, labeled by the elements of $[n] := \{1, \ldots, n\}$. Each vertex $i \in [n]$ has a community label $\sigma_i \in \{+1, -1\}$; these are drawn i.i.d. uniformly at random across all $i \in [n]$. Let $\boldsymbol{\sigma} := \{\sigma_i\}_{i=1}^n$ be the vector of community labels, with the two communities given by the sets $V_+ := \{i \in [n] : \sigma_i = +1\}$ and $V_- := \{i \in [n] : \sigma_i = -1\}$. Then, given the community labels $\boldsymbol{\sigma}$, the edges of $G$ are drawn independently across vertex pairs as follows. For distinct $i, j \in [n]$, if $\sigma_i \sigma_j = 1$, then the edge $(i, j)$ is in $G$ with probability $p$; else $(i, j)$ is in $G$ with probability $q$.

**Community recovery.** Generally speaking, a community recovery algorithm takes as input $G$ (without knowledge of the community labels $\boldsymbol{\sigma}$) and outputs a community labeling $\widehat{\boldsymbol{\sigma}}$. The *overlap* between the estimated labeling and the ground truth is given by

$$\mathsf{ov}(\widehat{\boldsymbol{\sigma}}, \boldsymbol{\sigma}) := \frac{1}{n} \left| \sum_{i=1}^n \widehat{\boldsymbol{\sigma}}_i \boldsymbol{\sigma}_i \right|.$$

In the formula for the overlap, we take an absolute value since the labelings $\boldsymbol{\sigma}$ and $-\boldsymbol{\sigma}$ specify the same community partition (and it is only possible to recover $\boldsymbol{\sigma}$ up to its sign). Moreover, notice that $\mathsf{ov}(\widehat{\boldsymbol{\sigma}}, \boldsymbol{\sigma})$ is always between 0 and 1, where a larger value corresponds to a better match between the estimated communities and the ground truth. Indeed, the algorithm succeeds in exactly recovering the communities (i.e., $\widehat{\boldsymbol{\sigma}} = \boldsymbol{\sigma}$ or $\widehat{\boldsymbol{\sigma}} = -\boldsymbol{\sigma}$) if and only if $\mathsf{ov}(\widehat{\boldsymbol{\sigma}}, \boldsymbol{\sigma}) = 1$.

In the logarithmic degree regime—that is, when $p = \alpha \log(n)/n$ and $q = \beta \log(n)/n$ for some fixed constants $\alpha, \beta \geq 0$—it is well-known that there is a sharp information-theoretic threshold for exactly recovering communities in the SBM [2, 45, 3, 1]. Specifically, if

$$\left| \sqrt{\alpha} - \sqrt{\beta} \right| > \sqrt{2}, \tag{2.1}$$

then exact recovery is possible: there is a polynomial-time algorithm which outputs an estimator $\widehat{\boldsymbol{\sigma}}$ satisfying $\lim_{n \to \infty} \mathbb{P}(\mathsf{ov}(\widehat{\boldsymbol{\sigma}}, \boldsymbol{\sigma}) = 1) = 1$. On the other hand, if

$$\left| \sqrt{\alpha} - \sqrt{\beta} \right| < \sqrt{2}, \tag{2.2}$$

then exact recovery is impossible: for *any* estimator $\widetilde{\boldsymbol{\sigma}}$, we have that $\lim_{n \to \infty} \mathbb{P}(\mathsf{ov}(\widetilde{\boldsymbol{\sigma}}, \boldsymbol{\sigma}) = 1) = 0$.

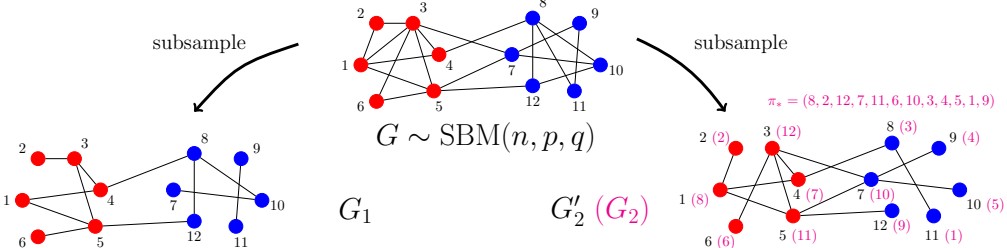

Figure 1: Schematic showing the construction of correlated SBMs (see text for details).

**Correlated SBMs.** The goal of our work is to understand how side information in the form of multiple *correlated* SBMs affects the threshold given by (2.1) and (2.2). To construct a pair of correlated SBMs, we define an additional parameter $s \in [0, 1]$ which controls the level of correlation between the two graphs. Formally, we construct $(G_1, G_2) \sim \mathrm{CSBM}(n, p, q, s)$ as follows. First, generate a parent graph $G \sim \mathrm{SBM}(n, p, q)$, and let $\boldsymbol{\sigma}$ denote the community labels. Next, given $G$, we construct $G_1$ and $G_2'$ by independent subsampling: each edge of $G$ is included in $G_1$ with probability $s$, independently of everything else, and non-edges of $G$ remain non-edges in $G_1$; we obtain $G_2'$ independently in the same fashion. Note that $G_1$ and $G_2'$ inherit the vertex labels from the parent graph $G$, and the community labels are given by $\boldsymbol{\sigma}$ in both graphs. Finally, we let $\pi_*$ be a uniformly random permutation of $[n]$, independently of everything else, and generate $G_2$ by relabeling the vertices of $G_2'$ according to $\pi_*$ (e.g., vertex $i$ in $G_2'$ is relabeled to $\pi_*(i)$ in $G_2$). This last step in the construction of $G_2$ reflects the observation that in applications, node labels are often obscured. This construction is visualized in Figure 1.

This model of correlated SBMs was first studied by Onaran, Erkip, and Garg [50]. This process of generating correlated graphs (i.e., by first generating a parent graph, independently subsampling it, and randomly permuting the labels) is a natural and common approach for inducing correlation in the formation of edges, and has been employed to study correlated graphs from the Erdős-Rényi model (see, e.g., [53], as well as further references in Section 4), the Chung-Lu model [62], and the preferential attachment model [35].

An important property of the construction is that marginally $G_1$ and $G_2$ are both SBMs. Specifically, since the subsampling probability is $s$, we have that $G_1 \sim \mathrm{SBM}(n, ps, qs)$. In the logarithmic degree regime, where $p = \alpha \log(n)/n$ and $q = \beta \log(n)/n$, (2.1) implies that the communities can be exactly recovered from $G_1$ *alone* if

$$\left| \sqrt{\alpha} - \sqrt{\beta} \right| > \sqrt{\frac{2}{s}}. \tag{2.3}$$

A central question of our work is how one can utilize the side information in $G_2$ to go *beyond* the single-graph threshold (2.3). This is formalized as follows.

**Objective 1** (Exact community recovery). *Given* $(G_1, G_2) \sim \mathrm{CSBM}\left(n, \frac{\alpha \log n}{n}, \frac{\beta \log n}{n}, s\right)$, *determine conditions on $\alpha$, $\beta$, and $s$ so that there exists an estimator $\widehat{\boldsymbol{\sigma}} = \widehat{\boldsymbol{\sigma}}(G_1, G_2)$ satisfying*

$$\lim_{n \to \infty} \mathbb{P}(\mathrm{ov}(\widehat{\boldsymbol{\sigma}}, \boldsymbol{\sigma}) = 1) = 1.$$

A key observation is that if the latent correspondence $\pi_*$ is known, then one can readily improve the achievability region in (2.3). Indeed, if $\pi_*$ is known, then one can reconstruct $G_2'$ from $G_2$. We can then construct a new graph $H_*$ by "overlaying" $G_1$ and $G_2'$ (i.e., taking their union). Formally, $(i, j)$ is an edge in $H_*$ if and only if $(i, j)$ is an edge in $G_1$ or $G_2'$. An equivalent interpretation is that $(i, j)$ is an edge in the parent graph $G$ and it is included in either $G_1$ or $G_2'$ in the subsampling process. The probability that the edge is *not* included in either $G_1$ or $G_2'$ is $(1 - s)^2$, so it follows that $H_* \sim \mathrm{SBM}\left(n, \alpha(1 - (1 - s)^2) \log(n)/n, \beta(1 - (1 - s)^2) \log(n)/n\right)$. By (2.1) it thus follows that exact community recovery is possible if

$$\left| \sqrt{\alpha} - \sqrt{\beta} \right| > \sqrt{\frac{2}{1 - (1 - s)^2}}. \tag{2.4}$$

Since $1 - (1-s)^2 > s$ for $s \in (0,1)$, (2.4) is a strict improvement over (2.3). Remarkably, this implies that if $\pi_*$ is known and if

$$\sqrt{\frac{2}{s}} > \left| \sqrt{\alpha} - \sqrt{\beta} \right| > \sqrt{\frac{2}{1 - (1-s)^2}},$$

then it is information-theoretically impossible to exactly recover $\boldsymbol{\sigma}$ from $G_1$ (or $G_2$) alone, but one can recover $\boldsymbol{\sigma}$ exactly by combining information from $G_1$ *and* $G_2$. To make this rigorous, we study when it is possible to exactly recover $\pi_*$ from $G_1$ and $G_2$. This task is known as *graph matching*.

**Objective 2** (Exact graph matching). *Given* $(G_1, G_2) \sim \mathrm{CSBM}\left(n, \frac{\alpha \log n}{n}, \frac{\beta \log n}{n}, s\right)$, *determine conditions on* $\alpha$, $\beta$, *and* $s$ *so that there exists an estimator* $\widehat{\pi} = \widehat{\pi}(G_1, G_2)$ *satisfying*

$$\lim_{n \to \infty} \mathbb{P}(\widehat{\pi} = \pi_*) = 1.$$

While we have motivated graph matching as an intermediate step in recovering communities, it is an important problem in its own right, with applications to data privacy in social networks [49, 53], protein-protein interaction networks [59], and machine learning [15, 14], among others. In particular, it is well known that graph matching algorithms can be used to de-anonymize social networks [49], showing that anonymity is not the same as privacy. Studying the fundamental limits of when graph matching is possible can serve to highlight the precise conditions when anonymity can indeed guarantee privacy, and when additional safeguards are necessary.

Although Objective 2 has not been studied previously, there is strong evidence of a phase transition for exact recovery of $\pi_*$ in the logarithmic degree regime. In the special case of correlated Erdős-Rényi graphs—that is, when $\alpha = \beta$—Cullina and Kiyavash [16, 17] showed that the maximum likelihood estimate exactly recovers $\pi_*$ with probability tending to 1 if $s^2 \alpha > 1$. When $\alpha \neq \beta$, and assuming that the community labels are *known* in both graphs, Onaran, Garg, and Erkip [50] showed that exact recovery of $\pi_*$ is possible if $s(1 - \sqrt{1 - s^2})(\alpha + \beta)/2 > 3$. Cullina et al. [19], also assuming that community labels are known in both graphs, stated (without proof) that exact recovery is possible if $s^2(\alpha + \beta)/2 > 2$. Since these works assume knowledge of community labels, it is unclear if these conditions allow to recover $\pi_*$ based on knowledge of only $G_1$ and $G_2$. Nevertheless, they suggest that exact graph matching may be possible in the logarithmic degree regime.

Turning to impossibility results, in correlated Erdős-Rényi graphs, if $s^2 \alpha < 1$, then there is no estimator which exactly recovers $\pi_*$ with probability bounded away from zero [16, 17, 61]. For correlated SBMs, Cullina et al. [19] showed that one cannot exactly recover $\pi_*$ when $s^2(\alpha + \beta)/2 < 1$.

In particular, for correlated Erdős-Rényi graphs the information-theoretic threshold $s^2 \alpha = 1$ is the *connectivity threshold* for the intersection graph of $G_1$ and $G_2'$. (Given two graphs $H_1$ and $H_2$, the edge $(i,j)$ is in the intersection graph of $H_1$ and $H_2$ if and only if it is an edge in both $H_1$ and $H_2$.) For correlated SBMs the connectivity threshold for the intersection graph is

$$s^2 \left( \frac{\alpha + \beta}{2} \right) = 1. \tag{2.5}$$

This suggests that (2.5) may be the information-theoretic threshold for exact recovery of $\pi_*$ for correlated SBMs. Our main result, Theorem 3.1, shows that this is indeed the case.

## 3 Results

We now describe our results, which address Objectives 1 and 2. In Section 3.1, we precisely characterize the fundamental information-theoretic limits for exact graph matching, thereby fully achieving Objective 2. In Section 3.2, we provide partial answers to Objective 1; in particular, these provide the information-theoretic threshold for exact community recovery in the regime where $s^2(\alpha + \beta)/2 > 1$. Finally, in Section 3.3, we extend the ideas of Section 3.2 to establish achievability and impossibility results for exact community recovery with $K$ correlated SBMs.

### 3.1 Exact Graph Matching

We start with our main result, which determines the achievability region for exact graph matching in correlated SBMs, providing an estimator that correctly recovers the latent vertex correspondence above the information-theoretic threshold.

**Theorem 3.1.** *Fix constants $\alpha, \beta > 0$ and $s \in [0, 1]$. Let $(G_1, G_2) \sim \text{CSBM}\left(n, \frac{\alpha \log n}{n}, \frac{\beta \log n}{n}, s\right)$. Let $\widehat{\pi}(G_1, G_2)$ be a vertex mapping that maximizes the number of agreeing edges between $G_1$ and $G_2$ (that is, the number of matched pairs of vertices for which an edge exists between them in both graphs). If*

$$s^2 \left(\frac{\alpha + \beta}{2}\right) > 1, \tag{3.1}$$

*then*

$$\lim_{n \to \infty} \mathbb{P}\left(\widehat{\pi}(G_1, G_2) = \pi_*\right) = 1.$$

We remark that the estimator $\widehat{\pi}$ used in Theorem 3.1 is a natural and well-motivated estimator for the latent mapping $\pi_*$. It was first considered by Pedarsani and Grossglauser [53] in the context of the correlated Erdős-Rényi model, where it is the maximum a posteriori (MAP) estimate [16, 17, 50]. As a result, it achieves the information-theoretic threshold for exact recovery of $\pi_*$ in the correlated Erdős-Rényi model [17, 61]. This estimator has also been studied in the context of correlated SBMs by Onaran, Erkip, and Garg [50]; they show that if the community labels of all vertices in $G_1$ and $G_2$ are known, then the permutation which maximizes the number of agreeing edges *and* is consistent with the community labels (i.e., does not map a vertex with label $+1$ to a vertex of label $-1$) succeeds in recovering $\pi_*$ exactly, provided that the (suboptimal) condition $s(1 - \sqrt{1 - s^2})(\alpha + \beta)/2 > 3$ holds. Theorem 3.1 improves on this result using a more refined analysis, obtaining the optimal condition (3.1), and not assuming any knowledge of community labels.

The next result establishes a converse to Theorem 3.1. This was previously proven in [19].

**Theorem 3.2.** *Fix constants $\alpha, \beta > 0$ and $s \in [0, 1]$. Let $(G_1, G_2) \sim \text{CSBM}\left(n, \frac{\alpha \log n}{n}, \frac{\beta \log n}{n}, s\right)$ and suppose that*

$$s^2 \left(\frac{\alpha + \beta}{2}\right) < 1. \tag{3.2}$$

*Then for any estimator $\widetilde{\pi}(G_1, G_2)$, we have that $\lim_{n \to \infty} \mathbb{P}(\widetilde{\pi}(G_1, G_2) = \pi_*) = 0$.*

Together, Theorems 3.1 and 3.2 establish the fundamental information-theoretic limits for exact recovery of $\pi_*$. This is the natural generalization of the corresponding results for correlated Erdős-Rényi graphs: when $\alpha = \beta$, the same estimator $\widehat{\pi}$ succeeds if $s^2 \alpha > 1$, else if $s^2 \alpha < 1$, then no estimator can exactly recover $\pi_*$ with probability bounded away from zero [17, 61].

An overview of the proofs of Theorems 3.1 and 3.2 is given in Section 5, with full details provided in the Supplementary Material.

## 3.2 Exact Community Recovery

We now turn to exact community recovery with two correlated SBMs, formalizing the arguments of Section 2. The strategy is to first perform exact graph matching, then to combine the two graphs by taking their union with respect to the matching, and finally to run an exact community recovery algorithm on this new graph.

**Theorem 3.3.** *Fix constants $\alpha, \beta > 0$ and $s \in [0, 1]$. Let $(G_1, G_2) \sim \text{CSBM}\left(n, \frac{\alpha \log n}{n}, \frac{\beta \log n}{n}, s\right)$. Suppose that $s^2(\alpha + \beta)/2 > 1$ and*

$$\left|\sqrt{\alpha} - \sqrt{\beta}\right| > \sqrt{\frac{2}{1 - (1 - s)^2}}. \tag{3.3}$$

*Then there is an estimator $\widehat{\boldsymbol{\sigma}} = \widehat{\boldsymbol{\sigma}}(G_1, G_2)$ such that*

$$\lim_{n \to \infty} \mathbb{P}\left(\text{ov}\left(\widehat{\boldsymbol{\sigma}}, \boldsymbol{\sigma}\right) = 1\right) = 1.$$

The proof readily follows from Theorem 3.1 and existing results on exact community recovery in the SBM [2, 45, 3, 1].

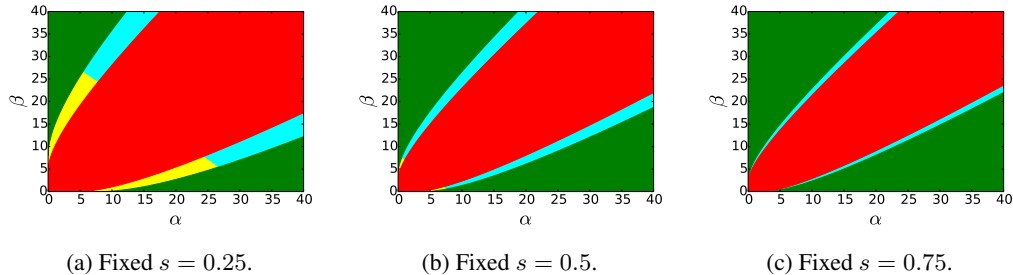

(a) Fixed $s = 0.25$.          (b) Fixed $s = 0.5$.          (c) Fixed $s = 0.75$.

Figure 2: Phase diagrams for exact community recovery for fixed $s$, with $\alpha \in [0, 40]$ and $\beta \in [0, 40]$ on the axes. *Green region:* exact community recovery is possible from $G_1$ alone; *Cyan region:* exact community recovery is impossible from $G_1$ alone, but it is possible from $(G_1, G_2)$; *Yellow region:* exact community recovery is impossible from $G_1$ alone, unknown if it is possible from $(G_1, G_2)$; *Red region:* exact community recovery is impossible from $(G_1, G_2)$.

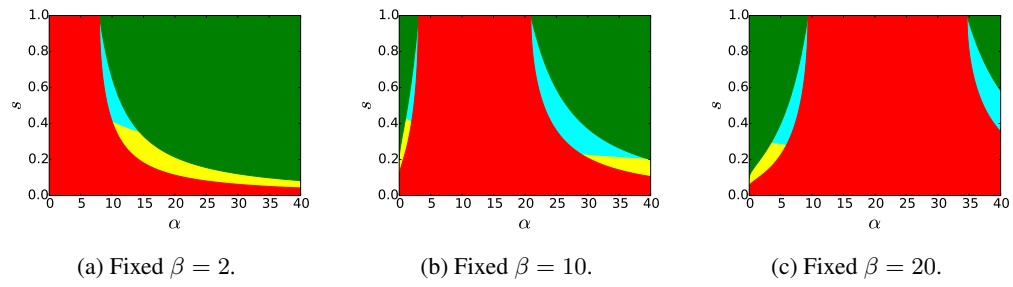

(a) Fixed $\beta = 2$.          (b) Fixed $\beta = 10$.          (c) Fixed $\beta = 20$.

Figure 3: Phase diagrams for exact community recovery for fixed $\beta$, with $\alpha \in [0, 40]$ and $s \in [0, 1]$ on the axes. (Colors as in Fig. 2.)

*Proof.* Given a permutation $\pi$ mapping $[n]$ to $[n]$, we let $G_1 \vee_\pi G_2$ be the *union graph with respect to $\pi$*, so that $(i, j)$ is an edge in $G_1 \vee_\pi G_2$ if and only if $(i, j)$ is an edge in $G_1$ or $(\pi(i), \pi(j))$ is an edge in $G_2$. In the special case where $\pi = \pi_*$, $H_* := G_1 \vee_{\pi_*} G_2$ is the subgraph of the parent graph $G$ consisting of edges that are in either $G_1$ or $G_2'$. It is readily seen that

$$H_* \sim \text{SBM}\left(n, \alpha(1 - (1-s)^2)\frac{\log n}{n}, \beta(1 - (1-s)^2)\frac{\log n}{n}\right). \tag{3.4}$$

The algorithm we study first computes $\widehat{\pi}(G_1, G_2)$ according to Theorem 3.1. We then pick any community recovery algorithm that is known to succeed until the information-theoretic limit, and run it on $\widehat{H} := G_1 \vee_{\widehat{\pi}} G_2$; we denote the result of this algorithm by $\widehat{\boldsymbol{\sigma}}(\widehat{H})$. We can then write

$$\mathbb{P}(\text{ov}(\widehat{\boldsymbol{\sigma}}(\widehat{H}), \boldsymbol{\sigma}) \neq 1) \leq \mathbb{P}(\{\text{ov}(\widehat{\boldsymbol{\sigma}}(\widehat{H}), \boldsymbol{\sigma}) \neq 1\} \cap \{\widehat{H} = H_*\}) + \mathbb{P}(\widehat{H} \neq H_*)$$
$$\leq \mathbb{P}(\text{ov}(\widehat{\boldsymbol{\sigma}}(H_*), \boldsymbol{\sigma}) \neq 1) + \mathbb{P}(\widehat{\pi} \neq \pi_\star),$$

where, to obtain the inequality in the second line, we have used that $\widehat{\boldsymbol{\sigma}}(\widehat{H}) = \widehat{\boldsymbol{\sigma}}(H_*)$ on the event $\{\widehat{H} = H_*\}$, and that $\widehat{H} \neq H_*$ implies $\widehat{\pi} \neq \pi_*$. Since exact community recovery on $H_*$ is possible when (3.3) holds [2, 45, 3, 1], we know that $\mathbb{P}(\text{ov}(\widehat{\boldsymbol{\sigma}}(H_*), \boldsymbol{\sigma}) \neq 1) \to 0$ as $n \to \infty$. In light of Theorem 3.1 we also have that $\mathbb{P}(\widehat{\pi} \neq \pi_*) \to 0$ when $s^2(\alpha + \beta)/2 > 1$, concluding the proof. $\square$

By the discussion in Section 2, Theorem 3.3 establishes the existence of a region of the parameter space where (i) there exists an algorithm that can exactly recover the communities using *both* $G_1$ and $G_2$, but (ii) it is information-theoretically impossible to do so using $G_1$ (or $G_2$) alone. Figures 2, 3, and 4 illustrate phase diagrams of the parameter space, where this region is highlighted in cyan.

To complement the achievability result of Theorem 3.3, our next result provides a condition under which exact community recovery is information-theoretically impossible.

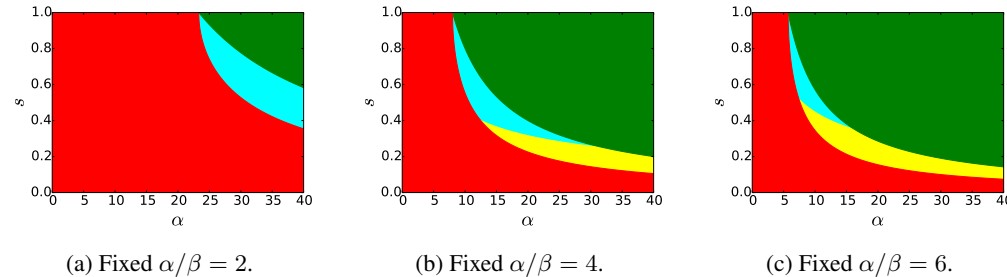

(a) Fixed $\alpha/\beta = 2$.      (b) Fixed $\alpha/\beta = 4$.      (c) Fixed $\alpha/\beta = 6$.

Figure 4: Phase diagrams for exact community recovery for fixed $\alpha/\beta$, with $\alpha \in [0, 40]$ and $s \in [0, 1]$ on the axes. (Colors as in Fig. 2.)

**Theorem 3.4.** *Fix constants $\alpha, \beta > 0$ and $s \in [0, 1]$. Let $(G_1, G_2) \sim \mathrm{CSBM}\left(n, \alpha\frac{\log n}{n}, \beta\frac{\log n}{n}, s\right)$ and suppose that*

$$\left|\sqrt{\alpha} - \sqrt{\beta}\right| < \sqrt{\frac{2}{1 - (1-s)^2}}. \tag{3.5}$$

*Then for any estimator $\widetilde{\boldsymbol{\sigma}} = \widetilde{\boldsymbol{\sigma}}(G_1, G_2)$, we have that $\lim_{n\to\infty} \mathbb{P}(\mathsf{ov}(\widetilde{\boldsymbol{\sigma}}, \boldsymbol{\sigma}) = 1) = 0$.*

The idea behind the proof is a simulation argument. Recall $H_* := G_1 \vee_{\pi_*} G_2$ from the proof of Theorem 3.3, and note that $H_* \sim \mathrm{SBM}\left(n, \alpha(1-(1-s)^2)\log(n)/n, \beta(1-(1-s)^2)\log(n)/n\right)$. From $H_*$ it is possible to simulate $(G_1, G_2)$, and so if exact community recovery is possible given $(G_1, G_2)$, then it is also possible given $H_*$. However, it is known [2, 45, 3, 1] that exact community recovery is not possible from $H_*$ if (3.5) holds. See Section D in the Supplementary Material for the full proof.

We remark that Theorem 3.4 provides a *partial converse* to the achievability result in Theorem 3.3: it is tight when $s^2(\alpha + \beta)/2 > 1$, but the precise information-theoretic threshold is unknown when $s^2(\alpha + \beta)/2 < 1$, which is the regime where exact graph matching fails. This leads to an interesting follow-up question: is exact graph matching *necessary* for the exact recovery of communities? We conjecture that it is not, which is formalized as follows.

**Conjecture 3.5.** *There exists $\epsilon = \epsilon(\alpha, \beta, s) > 0$ such that if (3.3) holds and*

$$s^2\left(\frac{\alpha + \beta}{2}\right) \geq 1 - \epsilon, \tag{3.6}$$

*then there is an estimator $\widehat{\boldsymbol{\sigma}} = \widehat{\boldsymbol{\sigma}}(G_1, G_2)$ such that $\lim_{n\to\infty} \mathbb{P}(\mathsf{ov}(\widehat{\boldsymbol{\sigma}}, \boldsymbol{\sigma}) = 1) = 1$.*

In words, we believe that the communities can be exactly recovered even in regimes where exact graph matching is information-theoretically impossible. We outline a possible way to prove this conjecture. The algorithm we shall use is the same one used in the proof of Theorem 3.3: we compute $\widehat{\pi}$, the permutation which maximizes the number of agreeing edges across $G_1$ and $G_2$, and then run an optimal community recovery algorithm on the union graph $\widehat{H} = G_1 \vee_{\widehat{\pi}} G_2$. Define the *correctly-matched region* $\mathcal{C} := \{i \in [n] : \widehat{\pi}(i) = \pi_*(i)\}$. When $s^2(\alpha + \beta)/2 < 1$, we have that $\mathcal{C} \neq [n]$ with high probability. However, we expect that $|\mathcal{C}| = (1 - o(1))n$; that is, $\widehat{\pi}$ coincides with $\pi_*$ on all but a negligible fraction of vertices (which is known as almost exact recovery). This is the case in correlated Erdős-Rényi graphs [18, 61], so we expect it to hold for correlated SBMs as well. Let $\widehat{H}_{\mathcal{C}}$ be the subgraph of $\widehat{H}$ restricted to the vertices in $\mathcal{C}$. Since all vertices in $\mathcal{C}$ have been correctly matched, we expect that (possibly in an *approximate* sense)

$$\widehat{H}_{\mathcal{C}} \sim \mathrm{SBM}\left(|\mathcal{C}|, \alpha(1-(1-s)^2)\frac{\log n}{n}, \beta(1-(1-s)^2)\frac{\log n}{n}\right). \tag{3.7}$$

In particular, if (3.3) holds, the communities of vertices in $\mathcal{C}$ can be *exactly* recovered. For vertices not in $\mathcal{C}$, note that most of the neighbors will be elements of $\mathcal{C}$, which will have correct community labels. If $\alpha > \beta$, then the true community label of a given vertex is the same as the true label of most neighbors with high probability (when $\alpha < \beta$, the reverse is true) [2], hence the community labels of vertices not in $\mathcal{C}$ can be correctly identified using a majority vote.

Making the arguments above formal is a challenging task. For one, though we may expect (3.7) to hold if $\mathcal{C}$ is a *fixed* set, it is in fact a *random* set depending on $G_1$, $G_2$, and $\pi_*$, so formally proving (3.7) requires a careful analysis. Moreover, we would like to use (3.7) to argue that running a community recovery algorithm on $\widehat{H}$ (rather than $\widehat{H}_\mathcal{C}$) perfectly recovers the communities in $\mathcal{C}$. Rigorously justifying these points requires significant effort, so we leave it to future work.

### 3.3 Multiple correlated stochastic block models

We next describe our results on how one can recover communities using $K$ correlated stochastic block models, again using graph matching as a subroutine. Considering more than two networks is more and more important in many applications, for instance in computational biology, where the increasing number of species for which protein-protein interaction networks are available can be leveraged for more powerful comparative studies [59, 34].

Formally, we construct $(G_1, \ldots, G_K) \sim \mathrm{CSBM}(n, p, q, s, K)$ as follows. First, generate a parent graph $G \sim \mathrm{SBM}(n, p, q)$, and let $\boldsymbol{\sigma}$ denote the community labels. Next, given $G$, we construct $G_1$ as well as $G'_2, \ldots, G'_K$ by independently subsampling $G$ with probability $s$. Finally, we let $\pi_*^2, \ldots, \pi_*^K$ be i.i.d. uniformly random permutations of $[n]$, independent of everything else, and for $2 \le k \le K$, we generate $G_k$ by relabeling the vertices of $G'_k$ according to $\pi_*^k$.

As in the case of two correlated graphs, the achievability and impossibility results depend on the structure of the union graph with respect to the true permutations $\pi_*^2, \ldots, \pi_*^K$.

**Theorem 3.6.** *Let* $(G_1, \ldots, G_K) \sim \mathrm{CSBM}\left(n, \frac{\alpha \log n}{n}, \frac{\beta \log n}{n}, s, K\right)$. *Suppose that* $s^2(\alpha + \beta)/2 > 1$ *and*

$$|\sqrt{\alpha} - \sqrt{\beta}| > \sqrt{\frac{2}{1 - (1-s)^K}}. \tag{3.8}$$

*Then there is an estimator* $\widehat{\boldsymbol{\sigma}} = \widehat{\boldsymbol{\sigma}}(G_1, \ldots, G_K)$ *such that* $\lim\limits_{n\to\infty} \mathbb{P}(\mathsf{ov}(\widehat{\boldsymbol{\sigma}}, \boldsymbol{\sigma}) = 1) = 1.$

Analogously to Theorem 3.3, Theorem 3.6 establishes the existence of a region of the parameter space where (i) there exists an algorithm that can exactly recover the communities using all of $G_1, G_2, \ldots, G_K$, but (ii) it is information-theoretically impossible to do so using only a strict subset of $G_1, G_2, \ldots, G_K$.

Our next result establishes an impossibility result which is analogous to Theorem 3.4.

**Theorem 3.7.** *Let* $(G_1, \ldots, G_K) \sim \mathrm{CSBM}\left(n, \alpha\frac{\log n}{n}, \beta\frac{\log n}{n}, s, K\right)$ *and suppose that*

$$|\sqrt{\alpha} - \sqrt{\beta}| < \sqrt{\frac{2}{1 - (1-s)^K}}. \tag{3.9}$$

*Then for any estimator* $\widetilde{\boldsymbol{\sigma}} = \widetilde{\boldsymbol{\sigma}}(G_1, G_2)$, *we have that* $\lim\limits_{n\to\infty} \mathbb{P}(\mathsf{ov}(\widetilde{\boldsymbol{\sigma}}, \boldsymbol{\sigma}) = 1) = 0.$

We highlight a few interesting aspects of Theorems 3.6 and 3.7. As in the two-graph case, Theorem 3.7 provides a *partial* converse to the achievability result in Theorem 3.6: it is tight in the regime $s^2(\alpha + \beta)/2 > 1$, but the correct threshold remains unknown when $s^2(\alpha + \beta)/2 < 1$. Additionally, as $K$ increases, the achievability and impossibility conditions in (3.8) and (3.9) converge to the conditions $|\sqrt{\alpha} - \sqrt{\beta}| > \sqrt{2}$ and $|\sqrt{\alpha} - \sqrt{\beta}| < \sqrt{2}$, which are the information-theoretic conditions for achievability and impossibility of community recovery in the *parent graph* $G$. In words, the more correlated graphs we observe, the less information is lost when generating the observed graphs from the parent graph via the sampling process.

## 4 Related work

Our work naturally draws upon techniques in the graph matching literature as well as the community recovery literature. Here, we elaborate on relevant work in these fields that were not covered during the exposition of our model and main results.

**Graph Matching.** Most of the theoretical literature on graph matching has focused on correlated Erdős-Rényi random graphs, which was introduced by Pedarsani and Grossglauser [53]. Significant progress has been made in recent years in characterizing the fundamental information-theoretic limits for recovering the latent vertex correspondence $\pi_*$. Cullina and Kiyavash [16, 17] first derived the precise information-theoretic conditions for *exact* recovery of $\pi_*$ for sparse graphs (in a sublinear-degree regime), and recently Wu, Xu, and Yu [61] refined this to include linear degree regimes. Our results, in particular Theorems 3.1 and 3.2, are the natural generalizations of these previous works to correlated SBMs, determining the precise information-theoretic threshold for exact recovery in this setting (and improving upon [50, 19]).

Weaker notions of recovery (e.g., almost exact recovery, partial recovery) have also been addressed for correlated Erdős-Rényi graphs (see [18, 27, 29, 28, 61] for more details). Recent work by Shirani, Erkip, and Garg [58] provides necessary and sufficient conditions for almost exact recovery in correlated SBMs. Our work is also a part of the growing literature studying correlated random graphs beyond the Erdős-Rényi model [50, 19, 35, 54, 58, 62].

A major open question is whether there exist *efficient algorithms* for inferring $\pi_*$ in correlated Erdős-Rényi graphs. In particular, the estimators which are known to succeed up to the information-theoretic threshold are usually given by the solution to a combinatorial optimization problem, for which a brute force search takes $O(n!)$ time. Significant improvements were recently made by [48, 6], who provided $n^{O(\log n)}$ time algorithms for exactly recovering $\pi_*$. For values of $s$ close to 1, recent work provides polynomial-time algorithms for exact recovery [23, 25, 40].

**Community Recovery in Multi-layer SBMs.** We briefly review the literature on multi-layer SBMs, as it is the form of side information studied in the literature that is closest to our work. Multi-layer SBMs were first introduced by Holland, Laskey, and Leinhardt, in their original work that introduced stochastic block models [31]. In this model, first a community labeling is chosen at random. Given the block structure, a collection of SBMs on the same vertex set with the same latent community labels are then generated, one for each layer, possibly with different (but known) edge formation probabilities. Variants of this model have been explored by several authors [30, 5, 51, 52, 36, 4, 7], but typically the layers are *conditionally independent* given the community labels. The works [43, 39] additionally consider node-level information that is correlated with the latent community membership. While our work also considers multiple networks as side information, we emphasize that there are significant differences. For one, the networks we consider are not conditionally independent given the latent communities, but are also correlated through the formation of edges. Moreover, in the multi-layer setting the node labels are known, which completely removes the need for graph matching.

## 5   Overview of graph matching proofs

**Achievability of exact graph matching: Proof sketch of Theorem 3.1.** Let $\mathcal{F}_\epsilon :=$ $\{(1-\epsilon)n/2 \leq |V_+|, |V_-| \leq (1+\epsilon)n/2\}$ denote the event that the two communities are approximately balanced. Since the community labels are i.i.d. uniform, we have for any fixed $\epsilon > 0$ that $\mathbb{P}(\mathcal{F}_\epsilon) = 1 - o(1)$ as $n \to \infty$; we may thus condition on $\mathcal{F}_\epsilon$. Let $S_{k_1,k_2}$ be the set of permutations which mismatches $k_1$ vertices in $V_+$ and $k_2$ vertices in $V_-$. We show that if (3.1) holds, then there exists $\epsilon = \epsilon(\alpha, \beta, s)$ sufficiently small so that

$$\mathbb{P}\left(\widehat{\pi} \in S_{k_1,k_2} \mid \mathcal{F}_\epsilon\right) \leq n^{-\epsilon(k_1+k_2)}. \tag{5.1}$$

To bound the probability that $\widehat{\pi} \neq \pi_*$, we then take a union bound over all the events $\{\widehat{\pi} \in S_{k_1,k_2}\}$ such that $k_1 + k_2 \geq 1$, that is, there is at least one mismatched vertex, concluding the proof.

The key technical result which enables the proof is (5.1); this is derived by deriving tight bounds for the generating function corresponding to the number of agreeing edges in $G_1$ and $G_2$ with respect to a given permutation. In prior work on the graph matching problem in correlated Erdős-Rényi graphs, as well as for correlated Gaussian matrices, the aforementioned generating functions could be exactly computed [16, 17, 61]. An important difference between work on these models and ours is that the stochastic block model is *heterogeneous*: the probability of edge formation is not i.i.d. over all vertex pairs, but varies depending on the latent community labels of the vertex pairs. As a result, the generating functions of interest cannot be explicitly computed. To handle this heterogeneity, we develop new techniques for bounding these generating functions. Specifically, we derive recursive bounds for the generating functions of interest as a function of the number of vertices; see Section C

in the Supplementary Material for details. We suspect that this method can be extended to analyze other classes of correlated networks with heterogeneous structure.

**Impossibility of exact graph matching: Proof sketch of Theorem 3.2.** Let $H$ be the intersection graph between $G_1$ and $G_2'$, that is, $(i, j)$ is an edge in $H$ if and only if $(i, j)$ is an edge in $G_1$ and $G_2'$. Equivalently, $(i, j)$ must be an edge in the parent graph $G$ and must be included in both $G_1$ and $G_2'$. Since the probability of the latter event is $s^2$, we see that $H \sim \text{SBM}\left(n, \alpha s^2 \log(n)/n, \beta s^2 \log(n)/n\right)$. If $s^2(\alpha + \beta)/2 < 1$, then $H$ is not connected with probability tending to 1 as $n \to \infty$. In particular, $H$ has many singletons in this regime, which are vertices that have *non-overlapping neighborhoods* in $G_1$ and $G_2'$. Due to the lack of shared information, it is difficult to match such vertices across the two graphs, even for optimal estimators that have access to the ground-truth community labeling $\boldsymbol{\sigma}$. In particular, one can show that the maximum a posteriori (MAP) estimator of $\pi_*$ given $G_1, G_2$, *and* $\boldsymbol{\sigma}$ cannot output $\pi_*$ with probability bounded away from zero, so neither can any other estimator.

# 6 Discussion and future work

In this work, we studied the problem of exact community recovery given multiple correlated SBMs as side information. Specifically, our goal was to understand how this side information changes the fundamental information-theoretic threshold for achievability and impossibility of exact community recovery. Strikingly, using multiple correlated SBMs allows one to exactly recover communities in regimes where it is information-theoretically impossible to do so using a single graph.

Precisely, we determine the sharp information-theoretic condition for exact graph matching in a pair of correlated SBMs. We then apply this to determine conditions for achievability and impossibility of exact community recovery. In the regime where exact graph matching is achievable, we identify the precise information-theoretic conditions for achievability and impossibility of exact community recovery. We also discuss extensions with $K \geq 2$ correlated SBMs.

Our work leaves open several important avenues for future work, which we outline below.

- **Closing the information-theoretic gaps in exact community recovery.** Together, Theorems 3.3 and 3.4 show that in the regime $s^2(\alpha + \beta)/2 > 1$, we have identified the information-theoretic threshold between impossibility and achievability for exact community recovery. However, we do not have achievability results for the regime $s^2(\alpha + \beta)/2 < 1$, since exact graph matching is not possible in this case. This leads to the following natural question which is formalized in Conjecture 3.5: *is exact graph matching needed for exact community recovery?* We believe the answer is *no*; we expect that showing this rigorously will lead to new algorithms for jointly synthesizing networks and identifying communities.

- **Efficient algorithms.** Our achievability algorithms rely on graph matching as a subroutine, which is computationally expensive. Do there exist *efficient* algorithms for graph matching in the correlated SBM model? If not, is it possible to recover communities exactly using a *polynomial-time relaxation* of the graph matching subroutine?

- **General correlated stochastic block models.** For simplicity of exposition, we focused on the simplest setting of the stochastic block model where there are two balanced communities. A natural future direction is to extend our results to account for more general SBMs with multiple communities (which are understood well in the single graph setting [1]).

- **Beyond exact community recovery.** Besides exact recovery, natural notions of community recovery include almost exact recovery, where the goal is to recover all but a negligible fraction of community labels, and partial recovery, where the goal is to do better than a random labeling. Using correlated networks as side information to accomplish these tasks is a natural and exciting direction. A key challenge is that in the regimes where phase transitions occur for almost exact and partial recovery (see [1]), exact graph matching is information-theoretically impossible by Theorem 3.2, hence this cannot be used as a black box. Solving this problem will lead to new methods for community detection based on data from multiple networks.

## Acknowledgments and Disclosure of Funding

We thank Jasmine Nirody for help with figures. We gratefully acknowledge funding support from NSF under Grant DMS 1811724 (to M.Z.R. and A.S.) and RAPID Grant IIS-2026982 (to A.S.).

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
