# Supplementary Material for
# Correlated Stochastic Block Models:
# Exact Graph Matching
# with Applications to Recovering Communities

**Miklós Z. Rácz**
Department of ORFE
Princeton University
Princeton, NJ 08544
mracz@princeton.edu

**Anirudh Sridhar**
Department of Electrical and Computer Engineering
Princeton University
Princeton, NJ 08544
anirudhs@princeton.edu

The supplementary material contains additional information on the information-theoretic thresholds for exact community recovery with correlated SBMs, as well as full proofs for the results in the main document. Sections A - E prove our main results in detail.

## A   Notation

Recall that the underlying vertex set is $V = [n] := \{1, 2, \ldots, n\}$. We denote by $\mathcal{S}_n$ the set of permutations of $[n]$. Recall that $V_+ := \{i \in [n] : \sigma_i = +1\}$ and $V_- := \{i \in [n] : \sigma_i = -1\}$ denote the vertices in the two communities.

Let $\mathcal{E} := \{\{i, j\} : i, j \in [n], i \neq j\}$ denote the set of all unordered vertex pairs. We will use $(i, j)$, $(j, i)$, and $\{i, j\}$ interchangeably to denote the unordered pair consisting of $i$ and $j$. Given $\boldsymbol{\sigma}$, we also define the sets $\mathcal{E}^+(\boldsymbol{\sigma}) := \{(i, j) \in \mathcal{E} : \sigma_i \sigma_j = +1\}$ and $\mathcal{E}^-(\boldsymbol{\sigma}) := \{(i, j) \in \mathcal{E} : \sigma_i \sigma_j = -1\}$. In words, $\mathcal{E}^+(\boldsymbol{\sigma})$ is the set of *intra-community* vertex pairs, and $\mathcal{E}^-(\boldsymbol{\sigma})$ is the set of *inter-community* vertex pairs. Note in particular that $\mathcal{E}^+(\boldsymbol{\sigma})$ and $\mathcal{E}^-(\boldsymbol{\sigma})$ partition $\mathcal{E}$.

We next introduce some notation pertaining to the construction of the correlated SBMs. Let $A$ be the adjacency matrix of $G_1$, let $B$ be the adjacency matrix of $G_2$, and let $B'$ be the adjacency matrix of $G_2'$. Note that, by construction, we have that $B'_{i,j} = B_{\pi_*(i), \pi_*(j)}$ for every $i, j$. By the construction of the correlated SBMs, we have the following probabilities for every $(i, j) \in \mathcal{E}$:

$$\mathbb{P}\left((A_{i,j}, B'_{i,j}) = (1,1) \,\middle|\, \boldsymbol{\sigma}\right) = \begin{cases} s^2 p & \text{if } \sigma_i = \sigma_j, \\ s^2 q & \text{if } \sigma_i \neq \sigma_j; \end{cases}$$

$$\mathbb{P}\left((A_{i,j}, B'_{i,j}) = (1,0) \,\middle|\, \boldsymbol{\sigma}\right) = \begin{cases} s(1-s)p & \text{if } \sigma_i = \sigma_j, \\ s(1-s)q & \text{if } \sigma_i \neq \sigma_j; \end{cases}$$

$$\mathbb{P}\left((A_{i,j}, B'_{i,j}) = (0,1) \,\middle|\, \boldsymbol{\sigma}\right) = \begin{cases} s(1-s)p & \text{if } \sigma_i = \sigma_j, \\ s(1-s)q & \text{if } \sigma_i \neq \sigma_j; \end{cases}$$

$$\mathbb{P}\left((A_{i,j}, B'_{i,j}) = (0,0) \,\middle|\, \boldsymbol{\sigma}\right) = \begin{cases} 1 - p(2s - s^2) & \text{if } \sigma_i = \sigma_j, \\ 1 - q(2s - s^2) & \text{if } \sigma_i \neq \sigma_j. \end{cases}$$

For brevity, for $i, j \in \{0, 1\}$ we write

$$p_{ij} := \mathbb{P}\left((A_{1,2}, B'_{1,2}) = (i, j) \,\middle|\, \boldsymbol{\sigma}\right) \qquad \text{if } \sigma_1 = \sigma_2$$

and

$$q_{ij} := \mathbb{P}\left((A_{1,2}, B'_{1,2}) = (i, j) \,\middle|\, \boldsymbol{\sigma}\right) \qquad \text{if } \sigma_1 \neq \sigma_2.$$

For an event $\mathcal{A}$, we denote by $\mathbf{1}(\mathcal{A})$ the indicator of $\mathcal{A}$, which is 1 if $\mathcal{A}$ occurs and 0 otherwise.

35th Conference on Neural Information Processing Systems (NeurIPS 2021).

## B    Exact graph matching for correlated SBMs: achievability

In this section we prove Theorem 3.1 in the main text. Recall that our objective is to find the ground truth permutation $\pi_*$. To this end, we study an estimator $\widehat{\pi}$ which maximizes the number of agreeing edges in the two graphs, that is, the number of pairs of vertices connected in both. In other words, letting $A$ denote the adjacency matrix of $G_1$ and $B$ denote the adjacency matrix of $G_2$, the estimator is given by

$$\widehat{\pi}(G_1, G_2) \in \arg\max_{\pi \in \mathcal{S}_n} \sum_{(i,j) \in \mathcal{E}} A_{i,j} B_{\pi(i), \pi(j)}. \tag{1}$$

When this estimator is not uniquely defined, that is, when the argmax set above is not a singleton, $\widehat{\pi}(G_1, G_2)$ is chosen to be an arbitrary element of the argmax set.

**Definition B.1** (Lifted permutation). *For a permutation $\pi \in \mathcal{S}_n$ on the vertices, define the corresponding* lifted permutation *$\tau : \mathcal{E} \to \mathcal{E}$ on vertex pairs as $\tau((i,j)) := (\pi(i), \pi(j))$. As a shorthand, we write $\tau = \ell(\pi)$, and thus also $\tau_* := \ell(\pi_*)$ and $\widehat{\tau} := \ell(\widehat{\pi})$.*

Note that if a permutation $\pi$ maps two vertices to each other, then the lifted permutation $\tau = \ell(\pi)$ maps this (unordered) pair of vertices to itself; that is, if $\pi(1) = 2$ and $\pi(2) = 1$, then $\tau((1,2)) = (2,1) = (1,2)$. Observe that there is a one-to-one mapping between permutations on vertices (i.e., $\mathcal{S}_n$) and lifted permutations. For this reason, finding the ground truth permutation $\pi_*$ is equivalent to finding the ground truth lifted permutation $\tau_*$. Similarly, conditioning on $\pi_*$ is equivalent to conditioning on $\tau_*$.

Using this notation, we can rewrite (1) as

$$\widehat{\pi}(G_1, G_2) \in \arg\max_{\pi \in \mathcal{S}_n} \sum_{e \in \mathcal{E}} A_e B_{\tau(e)}, \tag{2}$$

where $\tau = \ell(\pi)$, and $A_e = A_{i,j}$ if $e = (i,j)$. For a lifted permutation $\tau$ define

$$X(\tau) := \sum_{e \in \mathcal{E}} A_e B_{\tau_*(e)} - \sum_{e \in \mathcal{E}} A_e B_{\tau(e)} = \sum_{e \in \mathcal{E}: \tau(e) \neq \tau_*(e)} \left( A_e B_{\tau_*(e)} - A_e B_{\tau(e)} \right).$$

Observe that $X(\tau_*) = 0$ and that $\widehat{\pi}(G_1, G_2) \in \arg\min_{\pi \in \mathcal{S}_n} X(\ell(\pi))$. Therefore this estimator is correct—that is, $\widehat{\pi}(G_1, G_2) = \pi_*$—if for every lifted permutation $\tau \neq \tau_*$ we have that $X(\tau) > 0$. Conditioning on $\pi_*$ we thus have that

$$\mathbb{P}(\widehat{\pi} \neq \pi_*) \leq \mathbb{P}(\exists \pi \neq \pi_* : X(\ell(\pi)) \leq 0) = \mathbb{E}\left[ \mathbb{P}\left( \exists \pi \neq \pi_* : X(\ell(\pi)) \leq 0 \,|\, \pi_* \right) \right],$$

so a union bound implies that

$$\mathbb{P}(\widehat{\pi} \neq \pi_*) \leq \mathbb{E}\left[ \sum_{\pi \in \mathcal{S}_n : \pi \neq \pi_*} \mathbb{P}\left( X(\ell(\pi)) \leq 0 \,|\, \pi_* \right) \right].$$

To proceed, we shall bound the terms in the summation on the right hand side by studying the probability generating function (PGF) of $X(\tau)$ for any fixed lifted permutation $\tau$. More specifically, we will study the PGF of $X(\tau)$ given both $\pi_*$ (equivalently, $\tau_*$) and the community labeling $\boldsymbol{\sigma}$.

### B.1    Probabilistic bounds for $X(\tau)$

In this section, we establish large-deviations-type probability bounds for the event that $\widehat{\tau} = \tau$, where $\tau$ is a fixed lifted permutation. In our analysis we derive probability bounds which hold pointwise given *any* community labeling $\boldsymbol{\sigma}$ and ground truth lifted permutation $\tau_*$. We then derive simpler expressions for the bounds that hold when the two communities are approximately balanced.

To make these ideas more formal, we begin by defining some notation. Recall that given $\boldsymbol{\sigma}$, the set $\mathcal{E}^+(\boldsymbol{\sigma})$ is the set of intra-community vertex pairs, while $\mathcal{E}^-(\boldsymbol{\sigma})$ is the set of inter-community vertex

pairs. Given $\boldsymbol{\sigma}$ and $\tau_*$, for a fixed lifted permutation $\tau$ we also define the quantities

$$M^+(\tau) := \left|\left\{e \in \mathcal{E}^+(\boldsymbol{\sigma}) : \tau(e) \neq \tau_*(e)\right\}\right|,$$

$$M^-(\tau) := \left|\left\{e \in \mathcal{E}^-(\boldsymbol{\sigma}) : \tau(e) \neq \tau_*(e)\right\}\right|,$$

$$Y^+(\tau) := \sum_{e \in \mathcal{E}^+(\boldsymbol{\sigma}):\tau(e)\neq\tau_*(e)} A_e B_{\tau_*(e)},$$

$$Y^-(\tau) := \sum_{e \in \mathcal{E}^-(\boldsymbol{\sigma}):\tau(e)\neq\tau_*(e)} A_e B_{\tau_*(e)}.$$

In words, $M^+(\tau)$ is the number of mismatched intra-community vertex pairs. Furthermore, $Y^+(\tau)$ is the number of mismatched intra-community vertex pairs which contribute to the alignment score of the ground truth lifted permutation $\tau_* = \ell(\pi_*)$. We have analogous interpretations for the inter-community quantities $M^-(\tau)$ and $Y^-(\tau)$. Note that in addition to $\tau$, these quantities depend on $\boldsymbol{\sigma}$ and $\tau_*$ as well; however, we suppress this in the notation for simplicity. Observe also that $M^+(\tau)$ and $M^-(\tau)$ are deterministic functions of $\boldsymbol{\sigma}$, $\tau_*$, and $\tau$. On the other hand, given $\boldsymbol{\sigma}$ and $\tau_*$, and fixing $\tau$, the quantities $Y^+(\tau)$ and $Y^-(\tau)$ are random variables, since they depend on the two graphs $G_1$ and $G_2$ as well.

Given a community labeling $\boldsymbol{\sigma}$ and the ground truth lifted permutation $\tau_*$, for a fixed lifted permutation $\tau$ we shall study the PGF

$$\Phi^\tau(\theta, \omega, \zeta) := \mathbb{E}\left[\theta^{X(\tau)} \omega^{Y^+(\tau)} \zeta^{Y^-(\tau)} \,\Big|\, \boldsymbol{\sigma}, \tau_*\right].$$

**Remark B.2.** *Since our goal is to bound the probability of the event $\{X(\tau) \leq 0\}$, it is perhaps more natural to study simply the PGF of $X(\tau)$, rather than the joint PGF of $X(\tau)$, $Y^+(\tau)$, and $Y^-(\tau)$. However, the success of the former approach requires that $s^2(\alpha + \beta)/2 > 2$, which is suboptimal. For a tighter analysis, one must condition on the typical behavior of $Y^+(\tau)$ and $Y^-(\tau)$, which in turn requires us to consider the joint PGF. This idea was previously used to show that the information-theoretic threshold can be achieved in the graph matching problem for Erdős-Rényi graphs [6, 11].*

The next lemma provides a useful bound for $\Phi^\tau$ for any $\boldsymbol{\sigma}$ and $\tau_*$; we defer its proof to Section B.3.

**Lemma B.3.** *Given a community labeling $\boldsymbol{\sigma}$ and the ground truth lifted permutation $\tau_*$, the following holds. Fix $\pi \in \mathcal{S}_n$ and let $\tau = \ell(\pi)$. For any constants $\epsilon \in (0,1)$ and $1 \leq \omega, \zeta \leq 3$, it holds for all $n$ large enough that*

$$\Phi^\tau\left(1/\sqrt{n}, \omega, \zeta\right) \leq \exp\left(-(1-\epsilon)s^2\left(\alpha M^+(\tau) + \beta M^-(\tau)\right)\frac{\log n}{n}\right). \tag{3}$$

We remark that (3) bounds the probability generating function for the specific value $\theta = 1/\sqrt{n}$. This choice is somewhat arbitrary; the proof of Lemma B.3 shows that the bound holds for all $\theta$ smaller than some positive function of $\epsilon, \alpha, \beta$, and $s$, and larger than $\log(n)/n$. Similarly, the requirement that $\omega, \zeta \leq 3$ is arbitrary; we expect that, with a careful analysis, one could even let $\omega$ and $\zeta$ be slowly increasing functions of $n$.

To apply Lemma B.3 later on, we need to compute/estimate $M^+(\tau)$ and $M^-(\tau)$. To this end, given $\boldsymbol{\sigma}$ and $\pi_*$, for non-negative integers $k_1$ and $k_2$, let $S_{k_1,k_2}$ denote the set of lifted permutations $\ell(\pi)$ where $\pi$ incorrectly matches $k_1$ vertices in $V_+$ and incorrectly matches $k_2$ vertices in $V_-$. That is, define

$$S_{k_1,k_2} := \left\{\ell(\pi) : \left|\{i \in V_+ : \pi(i) \neq \pi_*(i)\}\right| = k_1 \text{ and } \left|\{i \in V_- : \pi(i) \neq \pi_*(i)\}\right| = k_2\right\}.$$

Note that $S_{k_1,k_2}$ is defined given $\boldsymbol{\sigma}$ and $\pi_*$; however, for simplicity we omit these from the notation. The next lemma employs simple counting arguments to compute $M^+(\tau)$ and $M^-(\tau)$ for $\tau \in S_{k_1,k_2}$. In essence, it shows how to go from mismatches in the vertex permutation $\pi$ to mismatches in the lifted permutation $\tau = \ell(\pi)$. We note that a variant of this result in a related but slightly different setting was stated (without proof) in [9]; we present the details for completeness.

**Lemma B.4.** *Fix $\pi \in \mathcal{S}_n$ and let $\tau = \ell(\pi)$. Given $\boldsymbol{\sigma}$ and $\pi_*$, let $k_1$ and $k_2$ be such that $\tau \in S_{k_1,k_2}$. Then we have that*

$$M^+(\tau) = \binom{k_1}{2} + k_1(|V_+| - k_1) + \binom{k_2}{2} + k_2(|V_-| - k_2) - \left|E_{tr}^+\right|; \tag{4}$$

$$M^-(\tau) = k_1|V_-| + k_2|V_+| - k_1 k_2 - \left|E_{tr}^-\right|, \tag{5}$$

*where*

$$E_{tr}^+ := \left\{ (u, v) \in \mathcal{E}^+(\boldsymbol{\sigma}) : \pi(u) = \pi_*(v), \pi(v) = \pi_*(u) \right\},$$

$$E_{tr}^- := \left\{ (u, v) \in \mathcal{E}^-(\boldsymbol{\sigma}) : \pi(u) = \pi_*(v), \pi(v) = \pi_*(u) \right\}.$$

*That is, $E_{tr}^+$ is the set of vertex pairs from the same community which are transposed under $\pi$ compared to $\pi_*$, and an analogous description holds for $E_{tr}^-$. Moreover, we have the bounds $\left| E_{tr}^+ \right|, \left| E_{tr}^- \right| \leq (k_1 + k_2)/2$.*

*Proof.* Let $e = (i, j)$. Observe first that if $\pi(i) = \pi_*(i)$ and $\pi(j) = \pi_*(j)$, then also $\tau(e) = \tau_*(e)$, and hence this pair does not contribute to $M^+(\tau)$ or $M^-(\tau)$. Thus in order for $e = (i, j)$ to contribute to $M^+(\tau)$ or $M^-(\tau)$, we must have either $\pi(i) \neq \pi_*(i)$ or $\pi(j) \neq \pi_*(j)$.

We start by deriving (4). Let us first consider the contribution to $M^+(\tau)$ from pairs of vertices in $V_+$. The number of pairs of vertices $i, j \in V_+$ such that $\pi(i) \neq \pi_*(i)$ and $\pi(j) \neq \pi_*(j)$ is $\binom{k_1}{2}$, while the number of pairs of vertices $i, j \in V_+$ such that one is correctly matched by $\pi$ and the other is incorrectly matched is $k_1(|V_+| - k_1)$. These give the first two terms in (4). However, not all of these pairs of vertices have $\tau(e) \neq \tau_*(e)$. Specifically, if $i, j \in V_+$ are such that $\pi(i) = \pi_*(j)$ and $\pi(j) = \pi_*(i)$, then both $i$ and $j$ are mismatched (and hence counted above), yet $\tau(e) = \tau_*(e)$ (and hence should not be counted). This leads to the subtraction in (4). The contribution to $M^+(\tau)$ from pairs in $V_-$ is analogous.

We now turn to deriving (5). The number of pairs where $i \in V_+$ and $j \in V_-$ such that $\pi(i) \neq \pi_*(i)$ is $k_1 |V_-|$. Similarly, the number of pairs where $i \in V_+$ and $j \in V_-$ such that $\pi(j) \neq \pi_*(j)$ is $k_2 |V_+|$. Here we have double-counted pairs $i \in V_+$ and $j \in V_-$ such that $\pi(i) \neq \pi_*(i)$ *and* $\pi(j) \neq \pi_*(j)$; there are $k_1 k_2$ such pairs. Thus the number of pairs $i \in V_+$ and $j \in V_-$ such that $\pi(i) \neq \pi_*(i)$ or $\pi(j) \neq \pi_*(j)$ is $k_1|V_-| + k_2|V_+| - k_1 k_2$. However, not all of these pairs of vertices have $\tau(e) \neq \tau_*(e)$. Specifically, if $i \in V_+$ and $j \in V_-$ are such that $\pi(i) = \pi_*(j)$ and $\pi(j) = \pi_*(i)$, then both $i$ and $j$ are mismatched (and hence counted above), yet $\tau(e) = \tau_*(e)$ (and hence should not be counted). This leads to the subtracted term in (5).

Finally, the total number of transpositions (of $\pi$ compared to $\pi_*$) satisfies $2 \left( \left| E_{tr}^+ \right| + \left| E_{tr}^- \right| \right) \leq k_1 + k_2$, since each transposition involves two mismatched vertices and $k_1 + k_2$ is the total number of mismatched vertices. This leads to the bounds $\left| E_{tr}^+ \right|, \left| E_{tr}^- \right| \leq (k_1 + k_2)/2$ as desired. $\qquad \square$

The combinatorial formulas (4) and (5) are somewhat unwieldy to use directly. Fortunately, we can derive relatively simple linear lower bounds when the two communities are approximately balanced. To formalize this idea, we first introduce the following "nice" event.

**Definition B.5** (Balanced communities). *For $\epsilon > 0$ define the event*

$$\mathcal{F}_\epsilon := \left\{ \left( 1 - \frac{\epsilon}{2} \right) \frac{n}{2} \leq |V_+|, |V_-| \leq \left( 1 + \frac{\epsilon}{2} \right) \frac{n}{2} \right\}.$$

Note that whether or not $\mathcal{F}_\epsilon$ holds depends only on the community labels $\boldsymbol{\sigma}$. Also, since the community labels are i.i.d. uniform, we have for any fixed $\epsilon > 0$ that $\mathbb{P}(\mathcal{F}_\epsilon) = 1 - o(1)$ as $n \to \infty$.

Now fix $\epsilon > 0$ and a lifted permutation $\tau$. Our next goal is to find simple lower bounds for $M^+(\tau)$ and $M^-(\tau)$, given community labels $\boldsymbol{\sigma}$ such that $\mathcal{F}_\epsilon$ holds, and given $\tau_*$. To this end, let $k_1$ and $k_2$ be such that $\tau \in S_{k_1, k_2}$. We distinguish two cases:

- *Case 1:* both $k_1$ and $k_2$ are small; specifically, $k_1 \leq \frac{\epsilon}{2}|V_+|$ and $k_2 \leq \frac{\epsilon}{2}|V_-|$.
- *Case 2:* either $k_1$ or $k_2$ is large; specifically, either $k_1 \geq \frac{\epsilon}{2}|V_+|$ or $k_2 \geq \frac{\epsilon}{2}|V_-|$.

We start with the first case, when $k_1 \leq \frac{\epsilon}{2}|V_+|$ and $k_2 \leq \frac{\epsilon}{2}|V_-|$.

**Lemma B.6.** *Fix $\epsilon > 0$. Given community labels $\boldsymbol{\sigma}$ such that $\mathcal{F}_\epsilon$ holds, let $k_1$ and $k_2$ be such that $k_1 \leq \frac{\epsilon}{2}|V_+|$ and $k_2 \leq \frac{\epsilon}{2}|V_-|$. Given $\boldsymbol{\sigma}$ and $\pi_*$, let $\tau$ be a lifted permutation such that $\tau \in S_{k_1, k_2}$. For all $n$ large enough we have the following bounds:*

$$M^+(\tau) \geq (1 - \epsilon)\frac{n}{2}(k_1 + k_2), \tag{6}$$

$$M^-(\tau) \geq (1 - \epsilon)\frac{n}{2}(k_1 + k_2). \tag{7}$$

*Proof.* For $n$ sufficiently large, we have the following lower bound for $M^+(\tau)$:

$$M^+(\tau) \overset{(a)}{\geq} k_1(|V_+| - k_1) + k_2(|V_-| - k_2) - \frac{k_1 + k_2}{2} \overset{(b)}{\geq} \left(1 - \frac{\epsilon}{2}\right)(k_1|V_+| + k_2|V_-|) - \frac{k_1 + k_2}{2}$$

$$\overset{(c)}{\geq} \left(\left(1 - \frac{\epsilon}{2}\right)^2 \frac{n}{2} - 1\right)(k_1 + k_2) \overset{(d)}{\geq} (1 - \epsilon)\frac{n}{2}(k_1 + k_2),$$

where $(a)$ follows from ignoring positive terms in the formula (4) and bounding $\left|E_{tr}^+\right|$ by $(k_1 + k_2)/2$, $(b)$ uses the upper bounds $k_1 \leq \frac{\epsilon}{2}|V_+|$ and $k_2 \leq \frac{\epsilon}{2}|V_-|$, $(c)$ uses the lower bounds $|V_+|, |V_-| \geq (1 - \epsilon/2)n/2$, which hold on the event $\mathcal{F}_\epsilon$, and finally $(d)$ uses $(1 - \epsilon/2)^2 > 1 - \epsilon$ and the fact that $n$ is sufficiently large. Turning to $M^-(\tau)$, we have the following lower bound:

$$M^-(\tau) \overset{(e)}{\geq} k_1|V_-| + k_2|V_+| - k_1 k_2 - \frac{k_1 + k_2}{2} = k_1\left(|V_-| - \frac{k_2}{2} - \frac{1}{2}\right) + k_2\left(|V_+| - \frac{k_1}{2} - \frac{1}{2}\right)$$

$$\overset{(f)}{\geq} \left(1 - \frac{\epsilon}{2}\right)(k_1|V_-| + k_2|V_+|) \overset{(g)}{\geq} \left(1 - \frac{\epsilon}{2}\right)^2 \frac{n}{2}(k_1 + k_2) \geq (1 - \epsilon)\frac{n}{2}(k_1 + k_2),$$

where $(e)$ follows from bounding $\left|E_{tr}^-\right|$ by $(k_1 + k_2)/2$ in the formula (5), $(f)$ uses $k_1 + 1 \leq \epsilon|V_+|$ and $k_2 + 1 \leq \epsilon|V_-|$, and finally $(g)$ uses the lower bounds $|V_+|, |V_-| \geq (1 - \epsilon/2)n/2$, which hold on the event $\mathcal{F}_\epsilon$. $\qquad\square$

Combining these estimates with Lemma B.3, the following lemma bounds the conditional probability that the estimate $\widehat{\pi}$ has $k_1$ mismatches in $V_1$ and $k_2$ mismatches in $V_2$, for small $k_1$ and $k_2$.

**Lemma B.7.** *Fix constants $\alpha, \beta > 0$, $s \in [0, 1]$, and $\epsilon \in (0, 1)$ such that $s^2(\alpha + \beta)/2 > (1 + \epsilon)(1 - \epsilon)^{-2}$. Given $\boldsymbol{\sigma}$, let $k_1$ and $k_2$ be such that $k_1 \leq \frac{\epsilon}{2}|V_+|$ and $k_2 \leq \frac{\epsilon}{2}|V_-|$. For all $n$ large enough we have that*

$$\mathbb{P}\left(\widehat{\tau} \in S_{k_1, k_2} \mid \boldsymbol{\sigma}, \tau_*\right) \mathbf{1}\left(\mathcal{F}_\epsilon\right) \leq n^{-\epsilon(k_1 + k_2)}. \tag{8}$$

*Proof.* Let $\tau \in S_{k_1, k_2}$. We then have that

$$\mathbb{P}\left(\widehat{\tau} = \tau \mid \boldsymbol{\sigma}, \tau_*\right) \overset{(a)}{\leq} \mathbb{P}\left(X(\tau) \leq 0 \mid \boldsymbol{\sigma}, \tau_*\right) = \mathbb{P}\left(n^{-X(\tau)/2} \geq 1 \mid \boldsymbol{\sigma}, \tau_*\right)$$

$$\overset{(b)}{\leq} \Phi^\tau\left(1/\sqrt{n}, 1, 1\right) \overset{(c)}{\leq} \exp\left(-(1 - \epsilon)s^2\left(\alpha M^+(\tau) + \beta M^-(\tau)\right)\frac{\log n}{n}\right),$$

where $(a)$ is due to the observation made earlier that $\widehat{\tau}$ is a minimizer of $X(\tau)$, and $X(\tau_*) = 0$; $(b)$ is due to Markov's inequality; and $(c)$ follows from Lemma B.3, for all $n$ large enough.

The estimate above allows us to bound the probability of interest via a union bound. To do this, we need to estimate $|S_{k_1, k_2}|$. Since there are $k_1 + k_2$ mismatched vertices in total, there are at most $\binom{n}{k_1 + k_2}$ ways to choose the set of mismatched vertices (this is a loose upper bound, since this formula disregards how many mismatched vertices there are of each community). The number of possible permutations on the mismatched vertices is at most $(k_1 + k_2)!$. Therefore

$$|S_{k_1, k_2}| \leq \binom{n}{k_1 + k_2}(k_1 + k_2)! = \frac{n!}{(n - k_1 - k_2)!} \leq n^{k_1 + k_2}.$$

Thus a union bound implies that

$$\mathbb{P}\left(\widehat{\tau} \in S_{k_1, k_2} \mid \boldsymbol{\sigma}, \tau_*\right) \leq |S_{k_1, k_2}| \max_{\tau \in S_{k_1, k_2}} \mathbb{P}\left(\widehat{\tau} = \tau \mid \boldsymbol{\sigma}, \tau_*\right)$$

$$\leq \max_{\tau \in S_{k_1, k_2}} \exp\left((k_1 + k_2)\log n - (1 - \epsilon)s^2\left(\alpha M^+(\tau) + \beta M^-(\tau)\right)\frac{\log n}{n}\right). \tag{9}$$

On the event $\mathcal{F}_\epsilon$, provided that $n$ is large enough, and $k_1 \leq \frac{\epsilon}{2}|V_+|$ and $k_2 \leq \frac{\epsilon}{2}|V_-|$, we may use the bounds in Lemma B.6 to bound the exponent in (9) from above by

$$\left\{1 - (1 - \epsilon)^2 s^2(\alpha + \beta)/2\right\}(k_1 + k_2)\log n \leq -\epsilon(k_1 + k_2)\log n,$$

where the second inequality follows from the assumption that $s^2(\alpha + \beta)/2 > (1 + \epsilon)(1 - \epsilon)^{-2}$. Plugging this into (9) we have thus obtained (8). $\qquad\square$

Next, we consider the second case, when either $k_1$ or $k_2$ is large; specifically, either $k_1 \geq \frac{\epsilon}{2}|V_+|$ or $k_2 \geq \frac{\epsilon}{2}|V_-|$. Our goal is to obtain lemmas analogous to Lemmas B.6 and B.7 in this case as well.

**Lemma B.8.** *Fix $\pi \in \mathcal{S}_n$ and let $\tau = \ell(\pi)$. Fix $\epsilon > 0$. Given community labels $\boldsymbol{\sigma}$ such that $\mathcal{F}_\epsilon$ holds, and given $\pi_*$, let $k_1$ and $k_2$ be such that $\tau \in S_{k_1, k_2}$. For all $n$ large enough we have the following bounds:*

$$M^+(\tau) \geq (1 - \epsilon)\frac{n}{4}(k_1 + k_2), \tag{10}$$

$$M^-(\tau) \geq (1 - \epsilon)\frac{n}{4}(k_1 + k_2). \tag{11}$$

*Proof.* On the event $\mathcal{F}_\epsilon$, we have that

$$M^+(\tau) \geq \binom{k_1}{2} + k_1(|V_+| - k_1) + \binom{k_2}{2} + k_2(|V_-| - k_2) - \frac{k_1 + k_2}{2}$$

$$= k_1\left(|V_+| - \frac{k_1 + 2}{2}\right) + k_2\left(|V_-| - \frac{k_2 + 2}{2}\right)$$

$$\overset{(h)}{\geq} \frac{1}{2}\left(k_1(|V_+| - 2) + k_2(|V_-| - 2)\right) \overset{(i)}{\geq} (1 - \epsilon)\frac{n}{4}(k_1 + k_2),$$

where $(h)$ is due to $k_1 \leq |V_+|$ and $k_2 \leq |V_-|$, and $(i)$ uses $|V_+| - 2 \geq (1 - \epsilon/2)|V_+|$ and $|V_-| - 2 \geq (1 - \epsilon/2)|V_-|$, as well as $|V_+|, |V_-| \geq (1 - \epsilon/2)n/2$, which all hold on the event $\mathcal{F}_\epsilon$ for all $n$ large enough. For $M^-(\tau)$, we can use identical arguments to obtain (11). $\square$

Note that Lemma B.8 makes no assumptions on $k_1$ or $k_2$; however, the obtained lower bounds are smaller by a factor of $1/2$ compared to the bounds obtained in Lemma B.6 when $k_1$ and $k_2$ are both small. The bounds in Lemma B.8 are used to obtain the following result, which is the analogue of Lemma B.7.

**Lemma B.9.** *Fix constants $\alpha, \beta > 0$, $s \in [0, 1]$, and $\epsilon \in (0, 1)$ such that $s^2(\alpha + \beta)/2 > (1 + \epsilon)(1 - \epsilon)^{-2}$. There exists $\delta = \delta(\alpha, \beta, s, \epsilon) > 0$ such that the following holds. Given $\boldsymbol{\sigma}$, let $k_1$ and $k_2$ be such that either $k_1 \geq \frac{\epsilon}{2}|V_+|$ or $k_2 \geq \frac{\epsilon}{2}|V_-|$. For all $n$ large enough we have that*

$$\mathbb{P}\left(\widehat{\tau} \in S_{k_1, k_2} \mid \boldsymbol{\sigma}, \tau_*\right)\mathbf{1}(\mathcal{F}_\epsilon) \leq n^{-\delta(k_1 + k_2)}. \tag{12}$$

Due to the additional factor of $1/2$ in the lower bounds for $M^+(\tau)$ and $M^-(\tau)$ in Lemma B.8 (compared to Lemma B.6), one could replicate the proof of Lemma B.7 to show that if $s^2(\alpha + \beta)/2 > 2$, then (12) holds for appropriate $\delta$. In order to prove an achievability result for the *correct* threshold $s^2(\alpha + \beta)/2 > 1$, we employ a more careful analysis in which we condition on typical values of $Y^+(\tau)$ and $Y^-(\tau)$. Similar ideas were used in previous work on achieving the information-theoretic threshold for exact recovery in correlated Erdős-Rényi graphs [6, 11]. Since the proof of Lemma B.9 is more involved, we defer it to Section B.4.

## B.2 Proof of Theorem 3.1

The proof of Theorem 3.1 now readily follows from Lemmas B.7 and B.9.

*Proof of Theorem 3.1.* By assumption we have that $s^2(\alpha + \beta)/2 > 1$. Let $\epsilon > 0$ be sufficiently small so that $s^2(\alpha + \beta)/2 > (1 + \epsilon)(1 - \epsilon)^{-2}$, and hence the conditions of Lemmas B.7 and B.9 are satisfied. Let $\delta$ be given by Lemma B.9 and let $\gamma := \min\{\epsilon, \delta\}$.

We first argue that we may assume that the event $\mathcal{F}_\epsilon$ holds. We have that

$$\mathbb{P}\left(\widehat{\pi} \neq \pi_*\right) = \mathbb{P}\left(\widehat{\tau} \neq \tau_*\right) = \mathbb{E}\left[\mathbb{P}\left(\widehat{\tau} \neq \tau_* \mid \boldsymbol{\sigma}, \tau_*\right)\right] \leq \mathbb{E}\left[\mathbb{P}\left(\widehat{\tau} \neq \tau_* \mid \boldsymbol{\sigma}, \tau_*\right)\mathbf{1}\left(\mathcal{F}_\epsilon\right)\right] + \mathbb{P}\left(\mathcal{F}_\epsilon^c\right).$$

Since the community labels are i.i.d. uniform, we have that $\mathbb{P}\left(\mathcal{F}_\epsilon^c\right) \to 0$ as $n \to \infty$, and thus it remains to be shown that $\mathbb{E}\left[\mathbb{P}\left(\widehat{\tau} \neq \tau_* \mid \boldsymbol{\sigma}, \tau_*\right)\mathbf{1}\left(\mathcal{F}_\epsilon\right)\right] \to 0$ as $n \to \infty$.

If $\widehat{\tau} \neq \tau_*$, then $\widehat{\pi}$ must have some incorrectly matched vertices (since $\widehat{\pi}$ is a permutation, it cannot have just a single mismatched vertex); in other words, we must have that $\widehat{\tau} \in S_{k_1, k_2}$ for some $k_1$ and $k_2$ satisfying $k_1 + k_2 \geq 2$. Thus by Lemmas B.7 and B.9 we have that

$$\mathbb{P}\left(\widehat{\tau} \neq \tau_* \mid \boldsymbol{\sigma}, \tau_*\right)\mathbf{1}\left(\mathcal{F}_\epsilon\right) = \sum_{k_1, k_2 : k_1 + k_2 \geq 2} \mathbb{P}\left(\widehat{\tau} \in S_{k_1, k_2} \mid \boldsymbol{\sigma}, \tau_*\right)\mathbf{1}\left(\mathcal{F}_\epsilon\right) \leq \sum_{k_1, k_2 : k_1 + k_2 \geq 2} n^{-\gamma(k_1 + k_2)}.$$

Note that there are $\ell + 1$ different pairs $(k_1, k_2)$ such that $k_1 + k_2 = \ell$. Therefore

$$\sum_{k_1, k_2 : k_1 + k_2 \geq 2} n^{-\gamma(k_1 + k_2)} \leq \sum_{\ell = 2}^{\infty} (\ell + 1) \, n^{-\gamma \ell} = n^{-2\gamma} \sum_{\ell = 0}^{\infty} (\ell + 3) \, n^{-\gamma \ell} \leq C n^{-2\gamma}$$

for some finite constant $C$ depending only on $\gamma$ (and hence only on $\alpha$, $\beta$, and $s$). Putting together the previous two displays and taking an expectation we obtain that

$$\mathbb{E}\left[\mathbb{P}\left(\widehat{\tau} \neq \tau_* \,|\, \boldsymbol{\sigma}, \tau_*\right) \mathbf{1}\left(\mathcal{F}_\epsilon\right)\right] \leq C n^{-2\gamma},$$

which concludes the proof. $\qquad\square$

### B.3  Generating function analysis: Proof of Lemma B.3

#### B.3.1  Cycle decomposition of the PGF

We begin by presenting a convenient representation of $X(\tau)$ as a sum of independent random variables (conditioned on $\boldsymbol{\sigma}$ and $\tau_*$), based on an appropriate cycle decomposition. Let $\mathcal{C}$ be the cycle decomposition of the lifted permutation $\tau_*^{-1} \circ \tau$, and note that the pairs for which $\tau_*(e) = \tau(e)$ are the fixed points of $\tau_*^{-1} \circ \tau$. We can then write

$$X(\tau) = \sum_{e \in \mathcal{E} : \tau(e) \neq \tau_*(e)} \left(A_e B_{\tau_*(e)} - A_e B_{\tau(e)}\right) = \sum_{C \in \mathcal{C} : |C| \geq 2} \sum_{e \in C} \left(A_e B_{\tau_*(e)} - A_e B_{\tau(e)}\right)$$

$$=: \sum_{C \in \mathcal{C} : |C| \geq 2} X_C(\tau).$$

Note that $\left(\tau_* \circ \tau_*^{-1} \circ \tau\right)(e) = \tau(e)$, and hence $\{\tau(e)\}_{e \in C} = \{\tau_*(e)\}_{e \in C}$. Therefore $X_C(\tau)$ is a function of $\left\{\left(A_e, B_{\tau_*(e)}\right)\right\}_{e \in C} = \{(A_e, B_e')\}_{e \in C}$. Given $\boldsymbol{\sigma}$ and $\tau_*$, these only depend on the entries of the adjacency matrix of the parent graph corresponding to pairs $e \in C$, as well as the sampling variables corresponding to pairs $e \in C$. Thus, due to the disjointness of cycles, the random variables $\{X_C(\tau)\}_{C \in \mathcal{C} : |C| \geq 2}$ are mutually independent (given $\boldsymbol{\sigma}$ and $\tau_*$). This implies, in particular, that for any $\theta \in \mathbb{R}$ we have that

$$\mathbb{E}\left[\theta^{X(\tau)} \,\Big|\, \boldsymbol{\sigma}, \tau_*\right] = \prod_{C \in \mathcal{C} : |C| \geq 2} \mathbb{E}\left[\theta^{X_C(\tau)} \,\Big|\, \boldsymbol{\sigma}, \tau_*\right].$$

A similar factorization holds for $\Phi^\tau$, which is the PGF of interest. First, define

$$Y_C^+(\tau) := \sum_{e \in C \cap \mathcal{E}^+(\boldsymbol{\sigma}) : \tau(e) \neq \tau_*(e)} A_e B_{\tau_*(e)},$$

$$Y_C^-(\tau) := \sum_{e \in C \cap \mathcal{E}^-(\boldsymbol{\sigma}) : \tau(e) \neq \tau_*(e)} A_e B_{\tau_*(e)}.$$

Again due to the disjointness of cycles, the triples $\left\{\left(X_C(\tau), Y_C^+(\tau), Y_C^-(\tau)\right)\right\}_{C \in \mathcal{C} : |C| \geq 2}$ are mutually independent (given $\boldsymbol{\sigma}$ and $\tau_*$), so we have the factorization

$$\Phi^\tau(\theta, \omega, \zeta) = \prod_{C \in \mathcal{C} : |C| \geq 2} \mathbb{E}\left[\theta^{X_C(\tau)} \omega^{Y_C^+(\tau)} \zeta^{Y_C^-(\tau)} \,\Big|\, \boldsymbol{\sigma}, \tau_*\right] =: \prod_{C \in \mathcal{C} : |C| \geq 2} \Phi_C^\tau(\theta, \omega, \zeta). \qquad (13)$$

Given the factorization in (13), a key intermediate goal is to bound $\Phi_C^\tau$ for $C$ such that $|C| \geq 2$. This is accomplished by the following lemma.

**Lemma B.10.** *Given $\boldsymbol{\sigma}$ and $\tau_*$, the following holds. Fix a lifted permutation $\tau$, and let $C$ be a cycle in $\tau_*^{-1} \circ \tau$ such that $|C| \geq 2$. Then for any constants $\epsilon \in (0, 1)$ and $1 \leq \omega, \zeta \leq 3$, it holds for all $n$ large enough that*

$$\Phi_C^\tau\left(\frac{1}{\sqrt{n}}, \omega, \zeta\right) \leq \exp\left(-(1 - \epsilon)s^2 \left(\alpha \left|C \cap \mathcal{E}^+(\boldsymbol{\sigma})\right| + \beta \left|C \cap \mathcal{E}^-(\boldsymbol{\sigma})\right|\right) \frac{\log n}{n}\right). \qquad (14)$$

The proof can be found in Section B.3.2. We remark that prior literature studying similar PGFs in different contexts (correlated Erdős-Rényi graphs or correlated Gaussian matrices) was able to derive *exact* expressions for the PGF of a cycle due to the i.i.d. structure of the model considered [5, 6, 11]. Deriving exact formulae for the PGF of a cycle in correlated stochastic block models is significantly more challenging due to the *heterogeneity* induced by different community labels in the cycle. Specifically, if the elements of the cycle are labelled differently, one obtains different formulae for the PGFs, even if the *number* of inter-community and intra-community edges within the cycle are the same. The proof of Lemma B.10 instead focuses on establishing simple, recursive bounds for the PGF, which ultimately leads to the right hand side in (14). We expect that this technique may be useful more generally in heterogeneous random graphs with independent structure, such as those generated from the Chung-Lu model [4].

We now prove Lemma B.3, which follows readily from Lemma B.10.

*Proof of Lemma B.3.* Using (13) and (14), we have the bound

$$\Phi^\tau(\theta, \omega, \zeta) \leq \exp\left(-(1-\epsilon)s^2 \frac{\log n}{n} \sum_{C \in \mathcal{C}: |C| \geq 2} \left(\alpha \left|C \cap \mathcal{E}^+(\boldsymbol{\sigma})\right| + \beta \left|C \cap \mathcal{E}^-(\boldsymbol{\sigma})\right|\right)\right).$$

Since the cycles of $\mathcal{C}$ partition $\mathcal{E} = \mathcal{E}^+(\boldsymbol{\sigma}) \cup \mathcal{E}^-(\boldsymbol{\sigma})$, we have that

$$\sum_{C \in \mathcal{C}: |C| \geq 2} \left|C \cap \mathcal{E}^+(\boldsymbol{\sigma})\right| = \left|\mathcal{E}^+(\boldsymbol{\sigma})\right| - \sum_{C \in \mathcal{C}: |C| = 1} \left|C \cap \mathcal{E}^+(\boldsymbol{\sigma})\right|$$

$$= \left|\{e \in \mathcal{E}^+(\boldsymbol{\sigma}) : \tau(e) \neq \tau_*(e)\}\right|$$

$$= M^+(\tau).$$

Similarly,

$$\sum_{C \in \mathcal{C}: |C| \geq 2} \left|C \cap \mathcal{E}^-(\boldsymbol{\sigma})\right| = M^-(\tau),$$

and the desired result immediately follows. $\qquad\square$

### B.3.2 Bounding the PGF of a cycle: Proof of Lemma B.10

In the following we assume that $\boldsymbol{\sigma}$ and $\tau_*$ are given. We also fix a lifted permutation $\tau$, as well as a cycle $C$ in $\tau_*^{-1} \circ \tau$ with $|C| \geq 2$. We enumerate the elements of $C$ by $e_1, \ldots, e_{|C|}$, where $\left(\tau_*^{-1} \circ \tau\right)(e_k) = e_{k+1}$ for every $k \in \{1, \ldots, |C| - 1\}$, and $\left(\tau_*^{-1} \circ \tau\right)(e_{|C|}) = e_1$. For convenience of notation, we also define $e_{|C|+1} := e_1$, so that $\left(\tau_*^{-1} \circ \tau\right)(e_k) = e_{k+1}$ for every $1 \leq k \leq |C|$. Observe that, by applying $\tau_*$ to both sides of this equality, we have that

$$\tau(e_k) = \left(\tau_* \circ \tau_*^{-1} \circ \tau\right)(e_k) = \tau_*(e_{k+1}). \tag{15}$$

Additionally, for $1 \leq k \leq |C|$, we set $\lambda_k := +1$ if $e_k \in \mathcal{E}^+(\boldsymbol{\sigma})$ and $\lambda_k := -1$ if $e_k \in \mathcal{E}^-(\boldsymbol{\sigma})$. Observe that for every $i, j \in \{0, 1\}$ and $1 \leq k \leq |C|$ we have that

$$\mathbb{P}\left(\left(A_{e_k}, B_{\tau_*(e_k)}\right) = (i, j) \,\big|\, \boldsymbol{\sigma}, \tau_*\right) = \mathbb{P}\left(\left(A_{e_k}, B'_{e_k}\right) = (i, j) \,\big|\, \boldsymbol{\sigma}, \tau_*\right) = \begin{cases} p_{ij} & \text{if } \lambda_k = +1, \\ q_{ij} & \text{if } \lambda_k = -1. \end{cases} \tag{16}$$

Moreover, note that, given $\boldsymbol{\sigma}$ and $\tau_*$, the random pairs $\left\{\left(A_{e_k}, B_{\tau_*(e_k)}\right)\right\}_{k=1}^{|C|} = \left\{\left(A_{e_k}, B'_{e_k}\right)\right\}_{k=1}^{|C|}$ are mutually independent. Next, for $1 \leq k \leq |C|$, define the random variables

$$X_k := \sum_{\ell=1}^{k} A_{e_\ell} B_{\tau_*(e_\ell)} - A_{e_\ell} B_{\tau(e_\ell)},$$

$$Y_k^+ := \sum_{\ell=1}^{k} \mathbf{1}\left(\lambda_\ell = +1\right) A_{e_\ell} B_{\tau_*(e_\ell)},$$

$$Y_k^- := \sum_{\ell=1}^{k} \mathbf{1}\left(\lambda_\ell = -1\right) A_{e_\ell} B_{\tau_*(e_\ell)}.$$

In particular, by construction we have that $X_{|C|} = X_C(\tau)$, $Y_{|C|}^+ = Y_C^+(\tau)$, and $Y_{|C|}^- = Y_C^-(\tau)$. Due to (15), as well as using $B_{\tau_*(e)} = B_e'$ for every $e \in \mathcal{E}$, we may also write these quantities as

$$X_k = \sum_{\ell=1}^k A_{e_\ell} B_{\tau_*(e_\ell)} - A_{e_\ell} B_{\tau_*(e_{\ell+1})} = \sum_{\ell=1}^k A_{e_\ell} B_{e_\ell}' - A_{e_\ell} B_{e_{\ell+1}}',$$

$$Y_k^+ = \sum_{\ell=1}^k \mathbf{1}\left(\lambda_\ell = +1\right) A_{e_\ell} B_{e_\ell}',$$

$$Y_k^- = \sum_{\ell=1}^k \mathbf{1}\left(\lambda_\ell = -1\right) A_{e_\ell} B_{e_\ell}'.$$

From the display above we also have that the increments satisfy

$$X_k - X_{k-1} = \begin{cases} 1 & \text{if } \left(A_{e_k}, B_{e_k}'\right) = (1,1), B_{e_{k+1}}' = 0, \\ -1 & \text{if } \left(A_{e_k}, B_{e_k}'\right) = (1,0), B_{e_{k+1}}' = 1, \\ 0 & \text{else;} \end{cases} \tag{17}$$

$$Y_k^+ - Y_{k-1}^+ = \begin{cases} 1 & \text{if } \lambda_k = +1 \text{ and } \left(A_{e_k}, B_{e_k}'\right) = (1,1), \\ 0 & \text{else;} \end{cases} \tag{18}$$

$$Y_k^- - Y_{k-1}^- = \begin{cases} 1 & \text{if } \lambda_k = -1 \text{ and } \left(A_{e_k}, B_{e_k}'\right) = (1,1), \\ 0 & \text{else.} \end{cases} \tag{19}$$

Note, in particular, that none of these increments depend on $A_{e_{k+1}}$. Next, for $1 \le k \le |C|$ and $i, j, m \in \{0,1\}$, define the PGF

$$\phi_{k,ij,m}(\theta, \omega, \zeta) := \mathbb{E}\left[\theta^{X_k} \omega^{Y_k^+} \zeta^{Y_k^-} \,\middle|\, \boldsymbol{\sigma}, \tau_*, \left(A_{e_1}, B_{e_1}'\right) = (i,j), B_{e_{k+1}}' = m\right],$$

where we note that $\phi_{|C|,ij,m}$ is only defined when $j = m$, since $e_{|C|+1} \equiv e_1$. The following proposition relates these PGFs to $\Phi_C^\tau$, which is the PGF of interest.

**Proposition B.11.** *Consider the setting described above. We then have that*

$$\Phi_C^\tau(\theta, \omega, \zeta) = \begin{cases} \displaystyle\sum_{i,j \in \{0,1\}} p_{ij} \phi_{|C|,ij,j}(\theta, \omega, \zeta) & \text{if } \lambda_1 = +1, \\ \displaystyle\sum_{i,j \in \{0,1\}} q_{ij} \phi_{|C|,ij,j}(\theta, \omega, \zeta) & \text{if } \lambda_1 = -1. \end{cases}$$

*Proof.* First, recall that $e_{|C|+1} \equiv e_1$, so conditioning on $\left(A_{e_1}, B_{e_1}'\right)$ is the same as conditioning on $\left(A_{e_1}, B_{e_1}'\right)$ and $B_{e_{|C|+1}}'$. The claim then follows from the definition of $\Phi_C^\tau$ by conditioning on $\left(A_{e_1}, B_{e_1}'\right)$ and recalling the probabilities in (16). $\qquad\square$

The usefulness of defining the PGFs $\{\phi_{k,ij,m}\}_{1 \le k \le |C|; i,j,m \in \{0,1\}}$ is that we can compute them recursively in $k$ in a straightforward manner. To see this, let $2 \le k \le |C| - 1$ and first consider the case of $m = 0$. By conditioning on $\left(A_{e_k}, B_{e_k}'\right)$ and using the tower rule, we have that

$$\phi_{k,ij,0}(\theta, \omega, \zeta) = \mathbb{E}_{\left(A_{e_k}, B_{e_k}'\right)} \left[\mathbb{E}\left[\theta^{X_{k-1}+(X_k - X_{k-1})} \omega^{Y_{k-1}^+ + (Y_k^+ - Y_{k-1}^+)} \zeta^{Y_{k-1}^- + (Y_k^- - Y_{k-1}^-)}\right.\right.$$

$$\left.\left.\middle|\, \boldsymbol{\sigma}, \tau_*, \left(A_{e_1}, B_{e_1}'\right) = (i,j), B_{e_{k+1}}' = 0, \left(A_{e_k}, B_{e_k}'\right)\right]\right].$$

With the additional conditioning on $\left(A_{e_k}, B_{e_k}'\right)$, the increments $(X_k - X_{k-1})$, $(Y_k^+ - Y_{k-1}^+)$, and $(Y_k^- - Y_{k-1}^-)$ are now deterministic. Indeed, from (17) we see that, since $B_{e_{k+1}}' = 0$, the increment $X_k - X_{k-1}$ is equal to 1 if $\left(A_{e_k}, B_{e_k}'\right) = (1,1)$, otherwise it is zero. Similar statements can be made about the other increments based on (18) and (19). Pulling the contributions from the increments out and putting everything together, we have that

$$\phi_{k,ij,0} = \begin{cases} (p_{00} + p_{10})\phi_{k-1,ij,0} + (p_{01} + p_{11}\theta\omega)\phi_{k-1,ij,1} & \text{if } \lambda_k = +1, \\ (q_{00} + q_{10})\phi_{k-1,ij,0} + (q_{01} + q_{11}\theta\zeta)\phi_{k-1,ij,1} & \text{if } \lambda_k = -1. \end{cases} \tag{20}$$

Repeating similar arguments for the case $m = 1$ gives the following recursion for $2 \le k \le |C| - 1$:

$$
\begin{pmatrix} \phi_{k,ij,0} \\ \phi_{k,ij,1} \end{pmatrix} = \begin{cases} \begin{pmatrix} p_{00} + p_{10} & p_{01} + p_{11}\theta\omega \\ p_{00} + p_{10}\theta^{-1} & p_{01} + p_{11}\omega \end{pmatrix} \begin{pmatrix} \phi_{k-1,ij,0} \\ \phi_{k-1,ij,1} \end{pmatrix} & \text{if } \lambda_k = +1, \\[2em] \begin{pmatrix} q_{00} + q_{10} & q_{01} + q_{11}\theta\zeta \\ q_{00} + q_{10}\theta^{-1} & q_{01} + q_{11}\zeta \end{pmatrix} \begin{pmatrix} \phi_{k-1,ij,0} \\ \phi_{k-1,ij,1} \end{pmatrix} & \text{if } \lambda_k = -1. \end{cases}
\tag{21}
$$

The appropriate part of the recursion also holds for $k = |C|$, noting that $\phi_{|C|,ij,m}$ is only defined for $j = m$. To complete the description of the recursion, we have to also give the initial conditions, which are the PGFs for $k = 1$. These can be computed as follows:

$$
\phi_{1,ij,m} = \begin{cases} \theta^{-1} & \text{if } (i,j) = (1,0), m = 1, \\ \theta\omega & \text{if } (i,j) = (1,1), m = 0, \lambda_1 = +1, \\ \theta\zeta & \text{if } (i,j) = (1,1), m = 0, \lambda_1 = -1, \\ \omega & \text{if } (i,j) = (1,1), m = 1, \lambda_1 = +1, \\ \zeta & \text{if } (i,j) = (1,1), m = 1, \lambda_1 = -1, \\ 1 & \text{else.} \end{cases}
\tag{22}
$$

To analyze the recursion (21), we first derive a useful relation between $\phi_{k,ij,1}$ and $\phi_{k,ij,0}$, as stated in the following lemma. Define

$$
R := \max\left\{ \frac{1 + 2s}{1 - s}\left(\frac{p_{10}}{p_{00}}\right), \frac{1 + 2s}{1 - s}\left(\frac{q_{10}}{q_{00}}\right) \right\},
\tag{23}
$$

and note that $R = O(\log(n)/n)$ for every fixed $s \in [0, 1]$. (Since $p_{10}$ and $q_{10}$ each contain a factor of $(1 - s)$, this holds also for $s = 1$.)

**Lemma B.12.** *Consider the setting described above. Then for any $2 \le k \le |C| - 1$, $i, j \in \{0, 1\}$, $1 \le \omega, \zeta \le 3$, and $\theta$ satisfying $0 < \theta \le 1 - R$, we have that*

$$
\phi_{k,ij,1} \le \left(1 + R\theta^{-1}\right)\phi_{k,ij,0}.
\tag{24}
$$

*Proof.* Our proof is by induction on $k$. We first check that the base case holds for all $i, j \in \{0, 1\}$. For $(i, j) = (0, 0)$ or $(i, j) = (0, 1)$, we may take the base case to be $k = 1$. Indeed, in these cases we have, from (22), that $\phi_{1,ij,0} = \phi_{1,ij,1} = 1$, so (24) holds. For $(i, j) = (1, 0)$ or $(i, j) = (1, 1)$, we shall take the base case to be $k = 2$.

Consider now the case of $(i, j) = (1, 0)$. From (21) and (22) we then have that

$$
\phi_{2,10,0} = \begin{cases} p_{00} + p_{10} + p_{01}\theta^{-1} + p_{11}\omega & \text{if } \lambda_2 = +1, \\ q_{00} + q_{10} + q_{01}\theta^{-1} + q_{11}\zeta & \text{if } \lambda_2 = -1; \end{cases}
\tag{25}
$$

$$
\phi_{2,10,1} = \begin{cases} p_{00} + p_{10}\theta^{-1} + p_{01}\theta^{-1} + p_{11}\theta^{-1}\omega & \text{if } \lambda_2 = +1, \\ q_{00} + q_{10}\theta^{-1} + q_{01}\theta^{-1} + q_{11}\theta^{-1}\zeta & \text{if } \lambda_2 = -1. \end{cases}
$$

First consider the case of $\lambda_2 = +1$. Using $\omega \le 3$, we have that $\phi_{2,10,1} \le p_{00} + p_{10}\theta^{-1} + p_{01}\theta^{-1} + 3p_{11}\theta^{-1}$. Now using $p_{11} = \frac{s}{1-s}p_{10}$ and simplifying, we have that $\phi_{2,10,1} \le p_{00} + p_{01}\theta^{-1} + \frac{1+2s}{1-s}p_{10}\theta^{-1}$. By expanding the product it can be verified that

$$
p_{00} + p_{01}\theta^{-1} + \frac{1 + 2s}{1 - s}p_{10}\theta^{-1} \le \left(1 + \frac{(1 + 2s)p_{10}}{(1 - s)p_{00}}\theta^{-1}\right)\phi_{2,10,0},
$$

which concludes the check of (24) in this case. Analogously, if $\lambda_2 = -1$, then

$$
\phi_{2,10,1} \le \left(1 + \frac{(1 + 2s)q_{10}}{(1 - s)q_{00}}\theta^{-1}\right)\phi_{2,10,0},
$$

concluding the check of the base case for $(i, j) = (1, 0)$.

Finally, consider the case of $(i, j) = (1, 1)$. From (21) and (22) we then have that

$$
\phi_{2,11,0} = \begin{cases} p_{00}\theta\omega + p_{10}\theta\omega + p_{01}\omega + p_{11}\theta\omega^2 & \text{if } \lambda_1 = \lambda_2 = +1, \\ q_{00}\theta\omega + q_{10}\theta\omega + q_{01}\omega + q_{11}\theta\zeta\omega & \text{if } \lambda_1 = +1, \lambda_2 = -1, \\ p_{00}\theta\zeta + p_{10}\theta\zeta + p_{01}\zeta + p_{11}\theta\zeta\omega & \text{if } \lambda_1 = -1, \lambda_2 = +1, \\ q_{00}\theta\zeta + q_{10}\theta\zeta + q_{01}\zeta + q_{11}\theta\zeta^2 & \text{if } \lambda_1 = \lambda_2 = -1; \end{cases} \tag{26}
$$

$$
\phi_{2,11,1} = \begin{cases} p_{00}\theta\omega + p_{10}\omega + p_{01}\omega + p_{11}\omega^2 & \text{if } \lambda_1 = \lambda_2 = +1, \\ q_{00}\theta\omega + q_{10}\omega + q_{01}\omega + q_{11}\zeta\omega & \text{if } \lambda_1 = +1, \lambda_2 = -1, \\ p_{00}\theta\zeta + p_{10}\zeta + p_{01}\zeta + p_{11}\zeta\omega & \text{if } \lambda_1 = -1, \lambda_2 = +1, \\ q_{00}\theta\zeta + q_{10}\zeta + q_{01}\zeta + q_{11}\zeta^2 & \text{if } \lambda_1 = \lambda_2 = -1. \end{cases}
$$

Now if $\lambda_1 = +1$, then we have that $\phi_{2,11,1} = \theta\omega\phi_{2,10,1}$ and $\phi_{2,11,0} = \theta\omega\phi_{2,10,0}$, so (24) follows from the previous paragraph. If $\lambda_1 = -1$, then we have that $\phi_{2,11,1} = \theta\zeta\phi_{2,10,1}$ and $\phi_{2,11,0} = \theta\zeta\phi_{2,10,0}$, so (24) again follows from the previous paragraph.

Now that we have fully checked all base cases, we turn to the inductive step. Suppose that $\lambda_k = +1$; the other case where $\lambda_k = -1$ is similar (with $\{p_{ij}\}$ replaced with $\{q_{ij}\}$ and $\omega$ replaced with $\zeta$). From the recursion (21), we have that (24) is equivalent to

$$
\left(p_{00} + p_{10}\theta^{-1}\right)\phi_{k-1,ij,0} + (p_{01} + p_{11}\omega)\phi_{k-1,ij,1}
$$
$$
\leq \left(1 + R\theta^{-1}\right)\left((p_{00} + p_{10})\phi_{k-1,ij,0} + (p_{01} + p_{11}\theta\omega)\phi_{k-1,ij,1}\right),
$$

which in turn is equivalent to

$$
\left(p_{01} + p_{11}\omega - \left(1 + R\theta^{-1}\right)(p_{01} + p_{11}\theta\omega)\right)\phi_{k-1,ij,1}
$$
$$
\leq \left(\left(1 + R\theta^{-1}\right)(p_{00} + p_{10}) - \left(p_{00} + p_{10}\theta^{-1}\right)\right)\phi_{k-1,ij,0}.
$$

Note that the coefficient on the left hand side satisfies

$$
p_{01} + p_{11}\omega - \left(1 + R\theta^{-1}\right)(p_{01} + p_{11}\theta\omega) \leq p_{11}\omega \leq 3p_{11},
$$

so it suffices to show that

$$
3p_{11}\phi_{k-1,ij,1} \leq \left(\left(1 + R\theta^{-1}\right)(p_{00} + p_{10}) - \left(p_{00} + p_{10}\theta^{-1}\right)\right)\phi_{k-1,ij,0}.
$$

By the induction hypothesis we have that $\phi_{k-1,ij,1} \leq \left(1 + R\theta^{-1}\right)\phi_{k-1,ij,0}$, so it suffices to show that

$$
3\left(1 + R\theta^{-1}\right)p_{11} \leq \left(1 + R\theta^{-1}\right)(p_{00} + p_{10}) - \left(p_{00} + p_{10}\theta^{-1}\right). \tag{27}
$$

The assumption $\theta \leq 1 - R$ implies that $1 + R\theta^{-1} \leq \theta^{-1}$. Using this and also that $p_{11} = \frac{s}{1-s}p_{10}$, we may bound the left hand side of (27) as follows:

$$
3\left(1 + R\theta^{-1}\right)p_{11} \leq \frac{3s}{1-s}p_{10}\theta^{-1} = \left(\frac{1+2s}{1-s} \cdot \frac{p_{10}}{p_{00}} \cdot p_{00} - p_{10}\right)\theta^{-1}
$$
$$
\leq (Rp_{00} - p_{10})\theta^{-1} = R\theta^{-1}p_{00} - p_{10}\theta^{-1}, \tag{28}
$$

where we also used the definition of $R$. The right hand side of (28) is at most the right hand side of (27), which concludes the proof. $\square$

We are now ready to put everything together to prove Lemma B.10.

*Proof of Lemma B.10.* Set $\theta := 1/\sqrt{n}$. Since $R$, as defined in (23), satisfies $R = O\left(\log(n)/n\right)$, the condition $0 < \theta \leq 1 - R$ of Lemma B.12 is satisfied for all $n$ large enough. Moreover, since $R\theta^{-1} = O\left(\log(n)/\sqrt{n}\right)$, we can make $R\theta^{-1}$ arbitrarily small for $n$ large enough. To simplify notation in what follows, we write

$$
C^+ := |\{1 \leq k \leq |C| : \lambda_k = +1\}| = \left|C \cap \mathcal{E}^+\left(\boldsymbol{\sigma}\right)\right|,
$$
$$
C^- := |\{1 \leq k \leq |C| : \lambda_k = -1\}| = \left|C \cap \mathcal{E}^-\left(\boldsymbol{\sigma}\right)\right|.
$$

Our goal is to bound $\Phi_C^\tau$. Due to Proposition B.11, it suffices to bound the PGFs $\phi_{|C|,ij,j}$ for $i, j \in \{0, 1\}$. To do this, we use the recursion (21), as well as Lemma B.12. Ideally, we would like

to present a streamlined argument that works for all $i, j \in \{0, 1\}$ simultaneously. However, there are minor differences in boundary cases for different values of $i, j \in \{0, 1\}$. Specifically, while the bound (24) in Lemma B.12 holds for all $2 \le k \le |C| - 1$ and all $i, j \in \{0, 1\}$, for $k = 1$ it only holds when $i = 0$ (see the beginning of the proof of Lemma B.12). Furthermore, $\phi_{|C|,ij,m}$ is only defined for $j = m$. For these reasons, we bound $\phi_{|C|,ij,j}$ separately for each $i, j \in \{0, 1\}$ (while minimizing repeated arguments).

We first consider the case of $(i, j) = (0, 0)$ and bound $\phi_{|C|,00,0}$. In this case the bound (24) in Lemma B.12 holds for all $1 \le k \le |C| - 1$. Noting that for $(i, j) = (0, 0)$ the recursion (20) holds for all $2 \le k \le |C|$, by plugging in (24) we obtain that the following holds for all $2 \le k \le |C|$:

$$
\phi_{k,00,0} \le \begin{cases} \left(p_{00} + p_{10} + \left(1 + R\theta^{-1}\right)(p_{01} + p_{11}\theta\omega)\right)\phi_{k-1,00,0} & \text{if } \lambda_k = +1, \\ \left(q_{00} + q_{10} + \left(1 + R\theta^{-1}\right)(q_{01} + q_{11}\theta\zeta)\right)\phi_{k-1,00,0} & \text{if } \lambda_k = -1. \end{cases} \tag{29}
$$

To simplify this recursion, first note that $p_{01} + p_{11}\theta\omega = (1 + o(1))p_{01}$ as $n \to \infty$, since $p_{01}$ and $p_{11}$ are on the same order, $\omega$ is bounded, and $\theta = o(1)$. Also recall that $R\theta^{-1} = o(1)$. Consequently we have that

$$
p_{00} + p_{10} + \left(1 + R\theta^{-1}\right)(p_{01} + p_{11}\theta\omega) = p_{00} + p_{10} + (1 + o(1))p_{01} = 1 - (1 + o(1))p_{11}
$$

as $n \to \infty$. Thus for any fixed $\epsilon \in (0, 1)$ we have, for all $n$ large enough, that

$$
p_{00} + p_{10} + \left(1 + R\theta^{-1}\right)(p_{01} + p_{11}\theta\omega) \le 1 - (1 - \epsilon)p_{11} \le \exp\left(-(1 - \epsilon)p_{11}\right),
$$

where we have used the inequality $1 + x \le \exp(x)$. Similarly we have that

$$
q_{00} + q_{10} + \left(1 + R\theta^{-1}\right)(q_{01} + q_{11}\theta\zeta) \le \exp\left(-(1 - \epsilon)q_{11}\right)
$$

for all $n$ large enough. Plugging these inequalities back into (29), for all $n$ large enough the following holds for all $2 \le k \le |C|$:

$$
\phi_{k,00,0} \le \begin{cases} \exp\left(-(1 - \epsilon)p_{11}\right)\phi_{k-1,00,0} & \text{if } \lambda_k = +1, \\ \exp\left(-(1 - \epsilon)q_{11}\right)\phi_{k-1,00,0} & \text{if } \lambda_k = -1. \end{cases} \tag{30}
$$

Iterating this inequality and noting that $\phi_{1,00,0} = 1$, we have thus obtained that

$$
\phi_{|C|,00,0} \le \begin{cases} \exp\left(-(1 - \epsilon)\left\{\left(C^+ - 1\right)p_{11} + C^- q_{11}\right\}\right) & \text{if } \lambda_1 = +1, \\ \exp\left(-(1 - \epsilon)\left\{C^+ p_{11} + \left(C^- - 1\right)q_{11}\right\}\right) & \text{if } \lambda_1 = -1. \end{cases} \tag{31}
$$

Next, we turn to the case of $(i, j) = (0, 1)$, with the goal of bounding $\phi_{|C|,01,1}$. First, we shall bound $\phi_{|C|-1,01,0}$. By the same arguments as before (using the recursion and Lemma B.12), we have that, for all $n$ large enough, the following holds for all $2 \le k \le |C| - 1$:

$$
\phi_{k,01,0} \le \begin{cases} \exp\left(-(1 - \epsilon)p_{11}\right)\phi_{k-1,01,0} & \text{if } \lambda_k = +1, \\ \exp\left(-(1 - \epsilon)q_{11}\right)\phi_{k-1,01,0} & \text{if } \lambda_k = -1. \end{cases} \tag{32}
$$

Iterating this inequality and noting that $\phi_{1,01,0} = 1$, we thus have that

$$
\phi_{|C|-1,01,0} \le \begin{cases} \exp\left(-(1 - \epsilon)\left\{\left(C^+ - 2\right)p_{11} + C^- q_{11}\right\}\right) & \text{if } \lambda_1 = \lambda_{|C|} = +1, \\ \exp\left(-(1 - \epsilon)\left\{\left(C^+ - 1\right)p_{11} + \left(C^- - 1\right)q_{11}\right\}\right) & \text{if } \lambda_1 \cdot \lambda_{|C|} = -1, \\ \exp\left(-(1 - \epsilon)\left\{C^+ p_{11} + \left(C^- - 2\right)q_{11}\right\}\right) & \text{if } \lambda_1 = \lambda_{|C|} = -1. \end{cases}
$$

Recall that $p_{11}, q_{11} = O\left(\log(n)/n\right)$, and so $\exp(p_{11}), \exp(q_{11}) = 1 + O\left(\log(n)/n\right)$. Therefore regardless of the value of $\lambda_{|C|}$, we have that

$$
\phi_{|C|-1,01,0} \le \left(1 + O\left(\tfrac{\log n}{n}\right)\right) \cdot \begin{cases} \exp\left(-(1 - \epsilon)\left\{\left(C^+ - 1\right)p_{11} + C^- q_{11}\right\}\right) & \text{if } \lambda_1 = +1, \\ \exp\left(-(1 - \epsilon)\left\{C^+ p_{11} + \left(C^- - 1\right)q_{11}\right\}\right) & \text{if } \lambda_1 = -1. \end{cases} \tag{33}
$$

Now turning to $\phi_{|C|,01,1}$, the recursion and Lemma B.12 together give that

$$
\phi_{|C|,01,1} \le \begin{cases} \left(p_{00} + p_{10}\theta^{-1} + \left(1 + R\theta^{-1}\right)(p_{01} + p_{11}\omega)\right)\phi_{|C|-1,01,0} & \text{if } \lambda_{|C|} = +1, \\ \left(q_{00} + q_{10}\theta^{-1} + \left(1 + R\theta^{-1}\right)(q_{01} + q_{11}\zeta)\right)\phi_{|C|-1,01,0} & \text{if } \lambda_{|C|} = -1. \end{cases}
$$

Recalling the values of the parameters in these coefficients, regardless of the value of $\lambda_{|C|}$ we have that

$$\phi_{|C|,01,1} \leq \left(1 + O\left(\tfrac{\log n}{\sqrt{n}}\right)\right) \phi_{|C|-1,01,0}. \tag{34}$$

Plugging this back into (33), we obtain that

$$\phi_{|C|,01,1} \leq \left(1 + O\left(\tfrac{\log n}{\sqrt{n}}\right)\right) \cdot \begin{cases} \exp\left(-(1-\epsilon)\left\{(C^+ - 1)\,p_{11} + C^-\,q_{11}\right\}\right) & \text{if } \lambda_1 = +1, \\ \exp\left(-(1-\epsilon)\left\{C^+ p_{11} + (C^- - 1)\,q_{11}\right\}\right) & \text{if } \lambda_1 = -1. \end{cases} \tag{35}$$

Next, we turn to the case of $(i,j) = (1,0)$, with the goal of bounding $\phi_{|C|,10,0}$. Note that in this case the bound in (24) only holds for $2 \leq k \leq |C| - 1$. By the same arguments as before (using the recursion and Lemma B.12), we have that, for all $n$ large enough, the following holds for all $3 \leq k \leq |C|$:

$$\phi_{k,10,0} \leq \begin{cases} \exp\left(-(1-\epsilon)p_{11}\right) \phi_{k-1,10,0} & \text{if } \lambda_k = +1, \\ \exp\left(-(1-\epsilon)q_{11}\right) \phi_{k-1,10,0} & \text{if } \lambda_k = -1. \end{cases} \tag{36}$$

Iterating this inequality gives that

$$\phi_{|C|,10,0} \leq \phi_{2,10,0} \cdot \begin{cases} \exp\left(-(1-\epsilon)\left\{(C^+ - 2)\,p_{11} + C^-\,q_{11}\right\}\right) & \text{if } \lambda_1 = \lambda_2 = +1, \\ \exp\left(-(1-\epsilon)\left\{(C^+ - 1)\,p_{11} + (C^- - 1)\,q_{11}\right\}\right) & \text{if } \lambda_1 \cdot \lambda_2 = -1, \\ \exp\left(-(1-\epsilon)\left\{C^+ p_{11} + (C^- - 2)\,q_{11}\right\}\right) & \text{if } \lambda_1 = \lambda_2 = -1. \end{cases}$$

From (25) we have that $\phi_{2,10,0} = 1 + O\left(\log(n)/\sqrt{n}\right)$, regardless of the value of $\lambda_2$. Using again that $\exp(p_{11}), \exp(q_{11}) = 1 + O\left(\log(n)/n\right)$, we thus have that

$$\phi_{|C|,10,0} \leq \left(1 + O\left(\tfrac{\log n}{\sqrt{n}}\right)\right) \cdot \begin{cases} \exp\left(-(1-\epsilon)\left\{(C^+ - 1)\,p_{11} + C^-\,q_{11}\right\}\right) & \text{if } \lambda_1 = +1, \\ \exp\left(-(1-\epsilon)\left\{C^+ p_{11} + (C^- - 1)\,q_{11}\right\}\right) & \text{if } \lambda_1 = -1. \end{cases} \tag{37}$$

Finally, we turn to the case of $(i,j) = (1,1)$, with the goal of bounding $\phi_{|C|,11,1}$. Similarly to (34), we have that

$$\phi_{|C|,11,1} \leq \left(1 + O\left(\tfrac{\log n}{\sqrt{n}}\right)\right) \phi_{|C|-1,11,0}, \tag{38}$$

and so in the following we bound $\phi_{|C|-1,11,0}$. By the recursion and Lemma B.12, we have that, for all $n$ large enough, the following holds for all $3 \leq k \leq |C| - 1$:

$$\phi_{k,11,0} \leq \begin{cases} \exp\left(-(1-\epsilon)p_{11}\right) \phi_{k-1,11,0} & \text{if } \lambda_k = +1, \\ \exp\left(-(1-\epsilon)q_{11}\right) \phi_{k-1,11,0} & \text{if } \lambda_k = -1. \end{cases} \tag{39}$$

Iterating this inequality gives that

$$\phi_{|C|-1,11,0}$$
$$\leq \phi_{2,11,0} \cdot \begin{cases} \exp\left(-(1-\epsilon)\left\{(C^+ - 3)\,p_{11} + C^-\,q_{11}\right\}\right) & \lambda_1 = \lambda_2 = \lambda_{|C|} = +1, \\ \exp\left(-(1-\epsilon)\left\{(C^+ - 2)\,p_{11} + (C^- - 1)\,q_{11}\right\}\right) & |\{i \in \{1,2,|C|\} : \lambda_i = +1\}| = 2, \\ \exp\left(-(1-\epsilon)\left\{(C^+ - 1)\,p_{11} + (C^- - 2)\,q_{11}\right\}\right) & |\{i \in \{1,2,|C|\} : \lambda_i = +1\}| = 1, \\ \exp\left(-(1-\epsilon)\left\{C^+ p_{11} + (C^- - 3)\,q_{11}\right\}\right) & \lambda_1 = \lambda_2 = \lambda_{|C|} = -1. \end{cases}$$

From (26) we have that $\phi_{2,11,0} = O\left(1/\sqrt{n}\right)$, regardless of the values of $\lambda_1$ and $\lambda_2$. Using again that $\exp(p_{11}), \exp(q_{11}) = 1 + O\left(\log(n)/n\right)$, we thus have that

$$\phi_{|C|-1,11,0} \leq O\left(\tfrac{1}{\sqrt{n}}\right) \cdot \begin{cases} \exp\left(-(1-\epsilon)\left\{(C^+ - 1)\,p_{11} + C^-\,q_{11}\right\}\right) & \text{if } \lambda_1 = +1, \\ \exp\left(-(1-\epsilon)\left\{C^+ p_{11} + (C^- - 1)\,q_{11}\right\}\right) & \text{if } \lambda_1 = -1. \end{cases}$$

Plugging this back into (38), we thus have that

$$\phi_{|C|,11,1} \leq O\left(\tfrac{1}{\sqrt{n}}\right) \cdot \begin{cases} \exp\left(-(1-\epsilon)\left\{(C^+ - 1)\,p_{11} + C^-\,q_{11}\right\}\right) & \text{if } \lambda_1 = +1, \\ \exp\left(-(1-\epsilon)\left\{C^+ p_{11} + (C^- - 1)\,q_{11}\right\}\right) & \text{if } \lambda_1 = -1. \end{cases} \tag{40}$$

We have now computed bounds for $\phi_{|C|,ij,j}$ for all $i,j \in \{0,1\}$, and so we are now ready to bound $\Phi_C^\tau$. Suppose that $\lambda_1 = +1$; the other case is analogous. By Proposition B.11 and the bounds

in (31), (35), (37), and (40), we have that

$$\Phi_C^\tau = p_{00}\phi_{|C|,00,0} + p_{01}\phi_{|C|,01,1} + p_{10}\phi_{|C|,10,0} + p_{11}\phi_{|C|,11,1}$$
$$\leq \exp\left(-(1-\epsilon)\left\{\left(C^+ - 1\right)p_{11} + C^- q_{11}\right\}\right)$$
$$\cdot \left\{p_{00} + \left(1 + O\left(\frac{\log n}{\sqrt{n}}\right)\right)(p_{01} + p_{10}) + O\left(\frac{1}{\sqrt{n}}\right)p_{11}\right\}.$$

Observe that

$$p_{00} + \left(1 + O\left(\frac{\log n}{\sqrt{n}}\right)\right)(p_{01} + p_{10}) + O\left(\frac{1}{\sqrt{n}}\right)p_{11} = 1 - p_{11} + O\left(\frac{\log^2(n)}{n^{3/2}}\right),$$

so for all $n$ large enough this is at most $1 - (1-\epsilon)p_{11} \leq \exp\left(-(1-\epsilon)p_{11}\right)$. Plugging this back into the inequality above, we obtain that

$$\Phi_C^\tau \leq \exp\left(-(1-\epsilon)\left\{C^+ p_{11} + C^- q_{11}\right\}\right).$$

Recalling the definitions of $p_{11}$ and $q_{11}$ shows that we have obtained the desired inequality. $\qquad\square$

### B.4 Proof of Lemma B.9

For any $t^+$ and $t^-$ (to be chosen later) we have that

$$\mathbb{P}\left(\hat{\tau} \in S_{k_1,k_2} \,\middle|\, \boldsymbol{\sigma}, \tau_*\right) \leq \mathbb{P}\left(\exists \tau \in S_{k_1,k_2} : X(\tau) \leq 0 \,\middle|\, \boldsymbol{\sigma}, \tau_*\right)$$
$$\leq \mathbb{P}\left(\exists \tau \in S_{k_1,k_2} : X(\tau) \leq 0, Y^+(\tau) \geq t^+, Y^-(\tau) \geq t^- \,\middle|\, \boldsymbol{\sigma}, \tau_*\right) \quad (41)$$
$$+ \mathbb{P}\left(\exists \tau \in S_{k_1,k_2} : Y^+(\tau) \leq t^+ \,\middle|\, \boldsymbol{\sigma}, \tau_*\right) \quad (42)$$
$$+ \mathbb{P}\left(\exists \tau \in S_{k_1,k_2} : Y^-(\tau) \leq t^- \,\middle|\, \boldsymbol{\sigma}, \tau_*\right). \quad (43)$$

In the following we bound from above each of these three terms, starting with (41). For any $\tau \in S_{k_1,k_2}$, and any $\theta \in (0,1]$ and $\omega, \zeta \geq 1$, we have that

$$\mathbb{P}\left(X(\tau) \leq 0, Y^+(\tau) \geq t^+, Y^-(\tau) \geq t^- \,\middle|\, \boldsymbol{\sigma}, \tau_*\right)$$
$$= \sum_{k \leq 0} \sum_{k^+ \geq t^+} \sum_{k^- \geq t^-} \mathbb{P}\left(\left(X(\tau), Y^+(\tau), Y^-(\tau)\right) = \left(k, k^+, k^-\right) \,\middle|\, \boldsymbol{\sigma}, \tau_*\right)$$
$$\leq \sum_{k=-\infty}^{\infty} \sum_{k^+=-\infty}^{\infty} \sum_{k^-=-\infty}^{\infty} \theta^k \omega^{k^+ - t^+} \zeta^{k^- - t^-} \mathbb{P}\left(\left(X(\tau), Y^+(\tau), Y^-(\tau)\right) = \left(k, k^+, k^-\right) \,\middle|\, \boldsymbol{\sigma}, \tau_*\right)$$
$$= \omega^{-t^+} \zeta^{-t^-} \Phi^\tau(\theta, \omega, \zeta).$$

By taking a union bound over $\tau \in S_{k_1,k_2}$ and setting $(\theta, \omega, \zeta) := (1/\sqrt{n}, e, e)$, we can thus bound the expression in (41) from above by

$$|S_{k_1,k_2}| \max_{\tau \in S_{k_1,k_2}} e^{-t^+ - t^-} \Phi^\tau\left(1/\sqrt{n}, e, e\right).$$

Using the estimate $|S_{k_1,k_2}| \leq n^{k_1+k_2}$ (see the proof of Lemma B.7) and also Lemma B.3, we thus have that

$$\mathbb{P}\left(\exists \tau \in S_{k_1,k_2} : X(\tau) \leq 0, Y^+(\tau) \geq t^+, Y^-(\tau) \geq t^- \,\middle|\, \boldsymbol{\sigma}, \tau_*\right)$$
$$\leq \max_{\tau \in S_{k_1,k_2}} \exp\left((k_1+k_2)\log n - \left(t^+ + t^- + (1-\epsilon)s^2\left(\alpha M^+(\tau) + \beta M^-(\tau)\right)\frac{\log n}{n}\right)\right).$$

Noting that we may choose $t^+$ and $t^-$ as functions of $\tau$, set

$$t^+ := (1-\epsilon)s^2 \alpha \frac{\log n}{n} M^+(\tau) \qquad \text{and} \qquad t^- := (1-\epsilon)s^2 \beta \frac{\log n}{n} M^-(\tau). \quad (44)$$

In this way the expression in (41) is bounded from above by

$$\max_{\tau \in S_{k_1,k_2}} \exp\left((k_1+k_2)\log n - 2(1-\epsilon)s^2\left(\alpha M^+(\tau) + \beta M^-(\tau)\right)\frac{\log n}{n}\right). \quad (45)$$

On the event $\mathcal{F}_\epsilon$, provided that $n$ is large enough, we may use the bounds in Lemma B.8 for $M^+(\tau)$ and $M^-(\tau)$ to bound the exponent in (45) from above by

$$\left\{1 - (1-\epsilon)^2 s^2 \left(\alpha + \beta\right)/2\right\}(k_1 + k_2)\log n \le -\epsilon(k_1 + k_2)\log n,$$

where the second inequality follows from the assumption that $s^2(\alpha+\beta)/2 > (1+\epsilon)(1-\epsilon)^{-2}$. We have thus obtained, for all $n$ large enough, that

$$\mathbb{P}\left(\exists \tau \in S_{k_1, k_2} : X(\tau) \le 0, Y^+(\tau) \ge t^+, Y^-(\tau) \ge t^- \,\big|\, \boldsymbol{\sigma}, \tau_*\right)\mathbf{1}\left(\mathcal{F}_\epsilon\right) \le n^{-\epsilon(k_1+k_2)}.$$

Next, we turn to bounding (42), and recall that we have set $t^+$ as in (44). We shall first relate $Y^+(\tau)$ to a similar quantity which depends only on the correctly matched region of the corresponding vertex permutation $\pi$. Formally, given $\pi_*$ (equivalently, $\tau_*$), define the sets

$$F(\pi) := \{v \in V : \pi(v) = \pi_*(v)\} \qquad \text{and} \qquad \binom{F(\pi)}{2} := \{\{u, v\} : u, v \in F(\pi), u \neq v\}.$$

In words, $F(\pi)$ is the set of correctly matched vertices according to $\pi$, and $\binom{F(\pi)}{2}$ is the set of unordered pairs in $F(\pi)$. We can then write

$$Y^+(\tau) = \sum_{e \in \mathcal{E}^+(\boldsymbol{\sigma}):\tau(e) \neq \tau_*(e)} A_e B_{\tau_*(e)} \overset{(a)}{=} \sum_{e \in \mathcal{E}^+(\boldsymbol{\sigma}) \setminus \left(\binom{F(\pi)}{2} \cup E_{tr}^+\right)} A_e B_{\tau_*(e)}$$

$$= \sum_{e \in \mathcal{E}^+(\boldsymbol{\sigma}) \setminus \binom{F(\pi)}{2}} A_e B_{\tau_*(e)} - \sum_{e \in E_{tr}^+} A_e B_{\tau_*(e)} \overset{(b)}{\ge} \sum_{e \in \mathcal{E}^+(\boldsymbol{\sigma}) \setminus \binom{F(\pi)}{2}} A_e B_{\tau_*(e)} - \frac{n}{2}.$$

Above, $(a)$ follows since $\tau(e) = \tau_*(e)$ if and only if either both endpoints of $e$ are fixed points of $\pi$ or the endpoints of $e$ are a transposition in $\pi$; and $(b)$ follows since $A_e B_{\tau_*(e)} \in \{0, 1\}$, so the second summation is at most $\left|E_{tr}^+\right| \le (k_1 + k_2)/2$ (by Lemma B.4), which in turn is at most $n/2$. Hence $Y^+(\tau) \le t^+$ implies that

$$\sum_{e \in \mathcal{E}^+(\boldsymbol{\sigma}) \setminus \binom{F(\pi)}{2}} A_e B_{\tau_*(e)} \le t^+ + \frac{n}{2}. \tag{46}$$

To abbreviate notation, for $F \subseteq V$ let $H_F := \mathcal{E}^+(\boldsymbol{\sigma}) \setminus \binom{F}{2}$ (where we suppress dependence on $\boldsymbol{\sigma}$ in the notation for simplicity). Noting that $M^+(\tau) \le \left|H_{F(\pi)}\right|$ and recalling the definition of $t^+$, (46) further implies that

$$\sum_{e \in H_{F(\pi)}} A_e B_{\tau_*(e)} \le (1-\epsilon)s^2\alpha \frac{\log n}{n}\left|H_{F(\pi)}\right| + \frac{n}{2}. \tag{47}$$

Importantly, note that (given $\boldsymbol{\sigma}$ and $\tau_*$) the sum in (47) depends on $\pi$ (equivalently, $\tau$) only through $F(\pi)$. The same holds for the right hand side of (47). Therefore if there exists $\tau \in S_{k_1, k_2}$ such that $Y^+(\tau) \le t^+$, then there exists $F \subseteq V$ such that $|V_+ \setminus F| = k_1$, $|V_- \setminus F| = k_2$, and the inequality

$$Z_F := \sum_{e \in H_F} A_e B_{\tau_*(e)} \le (1-\epsilon)s^2\alpha \frac{\log n}{n}\left|H_F\right| + \frac{n}{2}. \tag{48}$$

holds. Thus turning to (42), a union bound gives that

$$\mathbb{P}\left(\exists \tau \in S_{k_1, k_2} : Y^+(\tau) \le t^+ \,\big|\, \boldsymbol{\sigma}, \tau_*\right)$$

$$\le \mathbb{P}\left(\exists F \subseteq V \text{ with } |V_+ \setminus F| = k_1, |V_- \setminus F| = k_2 : Z_F \le (1-\epsilon)s^2\alpha \frac{\log n}{n}\left|H_F\right| + \frac{n}{2} \,\bigg|\, \boldsymbol{\sigma}, \tau_*\right)$$

$$\le \sum_{F \subseteq V : |V_+ \setminus F| = k_1, |V_- \setminus F| = k_2} \mathbb{P}\left(Z_F \le (1-\epsilon)s^2\alpha \frac{\log n}{n}\left|H_F\right| + \frac{n}{2} \,\bigg|\, \boldsymbol{\sigma}, \tau_*\right)$$

$$\le 2^n \max_{F \subseteq V : |V_+ \setminus F| = k_1, |V_- \setminus F| = k_2} \mathbb{P}\left(Z_F \le (1-\epsilon)s^2\alpha \frac{\log n}{n}\left|H_F\right| + \frac{n}{2} \,\bigg|\, \boldsymbol{\sigma}, \tau_*\right). \tag{49}$$

Before continuing, we make a brief remark about the purpose of the above computations. If we were to deal with $Y^+(\tau)$ directly, and take a union bound over all $\tau \in S_{k_1,k_2}$, we would gain a factor of $|S_{k_1,k_2}| \leq n^{k_1+k_2} = \exp(\Theta(n \log n))$ from the union bound, which would be too large for our purposes. This is why it is important to switch from $Y^+(\tau)$ to the sum in (48): it allows us to take a union bound over a much smaller set, resulting in a factor of only $2^n$, as in (49).

Continuing the proof, our goal is to bound the probability in (49). Notice that (conditioned on $\boldsymbol{\sigma}$ and $\tau_*$) for every $e \in \mathcal{E}^+(\boldsymbol{\sigma})$ we have that

$$A_e B_{\tau_*(e)} \sim \text{Bernoulli}\left(s^2 \alpha \frac{\log n}{n}\right),$$

and these random variables are (conditioned on $\boldsymbol{\sigma}$ and $\tau_*$) mutually independent across $e \in \mathcal{E}^+(\boldsymbol{\sigma})$. Hence (conditioned on $\boldsymbol{\sigma}$ and $\tau_*$) we have that $Z_F \sim \text{Bin}\left(|H_F|, s^2 \alpha \log(n)/n\right)$. In particular, note that $\mathbb{E}[Z_F \mid \boldsymbol{\sigma}, \tau_*] = |H_F| s^2 \alpha \log(n)/n$.

Note that for any $F \subseteq V$ such that $|V_+ \setminus F| = k_1$ and $|V_- \setminus F| = k_2$, and for any $\pi$ such that $\ell(\pi) \in S_{k_1,k_2}$, we have that $|H_F| \geq M^+(\ell(\pi))$. Therefore Lemma B.8 implies that $|H_F| \geq ((1-\epsilon)/4)(k_1+k_2)n$ for all $n$ large enough. Recall that we assume that either $k_1 \geq \frac{\epsilon}{2}|V_+|$ or $k_2 \geq \frac{\epsilon}{2}|V_-|$. Therefore on the event $\mathcal{F}_\epsilon$ we have that $k_1 + k_2 \geq (\epsilon/2)(1-\epsilon/2)n/2$. Thus on the event $\mathcal{F}_\epsilon$ we have that $|H_F| = \Omega(n^2)$. Hence on the event $\mathcal{F}_\epsilon$ we have, for all $n$ large enough, that

$$(1-\epsilon)s^2\alpha \frac{\log n}{n}|H_F| + \frac{n}{2} \leq (1-\epsilon/2)|H_F| s^2\alpha \frac{\log n}{n} = (1-\epsilon/2)\mathbb{E}[Z_F \mid \boldsymbol{\sigma}, \tau_*].$$

Thus for all $n$ large enough we have that

$$\mathbb{P}\left(Z_F \leq (1-\epsilon)s^2\alpha \frac{\log n}{n}|H_F| + \frac{n}{2}\,\middle|\,\boldsymbol{\sigma}, \tau_*\right)\mathbf{1}(\mathcal{F}_\epsilon)$$
$$\leq \mathbb{P}\left(Z_F \leq \left(1-\frac{\epsilon}{2}\right)\mathbb{E}[Z_F \mid \boldsymbol{\sigma}, \tau_*]\,\middle|\,\boldsymbol{\sigma}, \tau_*\right)\mathbf{1}(\mathcal{F}_\epsilon).$$

By Bernstein's inequality we have that

$$\mathbb{P}\left(Z_F \leq \left(1-\frac{\epsilon}{2}\right)\mathbb{E}[Z_F \mid \boldsymbol{\sigma}, \tau_*]\,\middle|\,\boldsymbol{\sigma}, \tau_*\right) \leq \exp\left(-\frac{\frac{\epsilon^2}{8}\mathbb{E}[Z_F \mid \boldsymbol{\sigma}, \tau_*]^2}{\text{Var}(Z_F \mid \boldsymbol{\sigma}, \tau*) + \frac{1}{3}\mathbb{E}[Z_F \mid \boldsymbol{\sigma}, \tau_*]}\right)$$
$$\leq \exp\left(-\frac{3\epsilon^2}{32}\mathbb{E}[Z_F \mid \boldsymbol{\sigma}, \tau_*]\right),$$

where the second inequality uses the fact that $\text{Var}(Z_F \mid \boldsymbol{\sigma}, \tau_*) \leq \mathbb{E}[Z_F \mid \boldsymbol{\sigma}, \tau_*]$. Recall that

$$\mathbb{E}[Z_F \mid \boldsymbol{\sigma}, \tau_*] = |H_F| s^2\alpha \frac{\log n}{n} \geq \frac{1-\epsilon}{4}(k_1+k_2)s^2\alpha \log n$$

for all $n$ large enough. Putting everything together, we have thus shown, for any $F \subseteq V$ such that $|V_+ \setminus F| = k_1$ and $|V_- \setminus F| = k_2$, that

$$\mathbb{P}\left(Z_F \leq (1-\epsilon)s^2\alpha \frac{\log n}{n}|H_F| + \frac{n}{2}\,\middle|\,\boldsymbol{\sigma}, \tau_*\right)\mathbf{1}(\mathcal{F}_\epsilon) \leq \exp\left(-\frac{3\epsilon^2(1-\epsilon)}{128}(k_1+k_2)s^2\alpha \log n\right)$$

for all $n$ large enough. Plugging this back into (49), we obtain that

$$\mathbb{P}\left(\exists \tau \in S_{k_1,k_2} : Y^+(\tau) \leq t^+ \,\middle|\, \boldsymbol{\sigma}, \tau_*\right)\mathbf{1}(\mathcal{F}_\epsilon) \leq \exp\left(n \log 2 - \frac{3\epsilon^2(1-\epsilon)}{128}(k_1+k_2)s^2\alpha \log n\right)$$
$$\leq \exp\left(-\frac{\epsilon^2(1-\epsilon)}{50}(k_1+k_2)s^2\alpha \log n\right)$$

for all $n$ large enough, where the second inequality follows because $(k_1+k_2)\log n = \Omega(n \log n)$, which is asymptotically much larger than $n \log 2$. This concludes the bound for (42).

Turning to (43), repeating identical steps as above also shows, for all $n$ large enough, that

$$\mathbb{P}\left(\exists \tau \in S_{k_1,k_2} : Y^-(\tau) \leq t^- \,\middle|\, \boldsymbol{\sigma}, \tau_*\right)\mathbf{1}(\mathcal{F}_\epsilon) \leq \exp\left(-\frac{\epsilon^2(1-\epsilon)}{50}(k_1+k_2)s^2\beta \log n\right).$$

Putting everything together, if we let $\delta_0 := \min\left\{\epsilon, \epsilon^2(1-\epsilon)s^2\alpha/50, \epsilon^2(1-\epsilon)s^2\beta/50\right\}$, then the terms (41), (42), and (43) are all at most $n^{-\delta_0(k_1+k_2)}$, for all $n$ large enough. This gives a total bound of $3n^{-\delta_0(k_1+k_2)} \leq n^{-\delta_0(k_1+k_2)/2}$ for all $n$ large enough, concluding the proof of Lemma B.9.

# C   Exact graph matching for correlated SBMs: impossibility

In this section we prove Theorem 3.2 in the main text, showing that it is impossible to exactly match the two correlated SBMs $G_1$ and $G_2$ whenever $s^2(\alpha + \beta)/2 < 1$. While this was previously proven in [7], we provide a proof for completeness. At a high level, the strategy behind the proof is as follows.

When $s^2(\alpha + \beta)/2 < 1$, we show that there are many vertices in $G$ such that the corresponding vertices in $G_1$ and $G_2'$ have non-overlapping neighborhoods. Due to this lack of shared information, such vertices are challenging to correctly match in the two graphs, even for the maximum a posteriori (MAP) estimator that is given $G_1$ and $G_2$. For this reason, the MAP estimator is likely to output an incorrect vertex correspondence. Since the MAP estimator minimizes the probability of error, we conclude that no other estimator can do better (in particular, no estimator can output the correct correspondence with probability bounded away from zero).

The input to the estimation problem is the pair of labeled graphs $G_1$ and $G_2$; equivalently, in the following we use the respective adjacency matrices $A$ and $B$. To compute the MAP estimator, we need to derive the posterior distribution of $\pi_*$ given $A$ and $B$. This is unfortunately quite challenging in correlated SBMs, since the probability of edge formation depends on the (unknown) latent community memberships of vertices. To carry out a tractable analysis, we shall provide extra information to the estimator: we assume that $\boldsymbol{\sigma}$ is also known; that is, we assume knowledge of the community memberships of all vertices in $G_1$. Providing this extra information can only make the problem of estimating $\pi_*$ easier, yet it turns out that recovering $\pi_*$ is still impossible even with this extra information.

## C.1   Properties of the posterior distribution

Before deriving the posterior distribution of $\pi_*$ given $A$, $B$, and $\boldsymbol{\sigma}$, we define some relevant notation. Given $\boldsymbol{\sigma}$, for a lifted permutation $\tau$ and $i, j \in \{0, 1\}$, define

$$\mu^+(\tau)_{ij} := \sum_{e \in \mathcal{E}^+(\boldsymbol{\sigma})} \mathbf{1}\left(\left(A_e, B_{\tau(e)}\right) = (i, j)\right),$$

$$\mu^-(\tau)_{ij} := \sum_{e \in \mathcal{E}^-(\boldsymbol{\sigma})} \mathbf{1}\left(\left(A_e, B_{\tau(e)}\right) = (i, j)\right).$$

Additionally define

$$\nu^+(\tau) := \sum_{e \in \mathcal{E}^+(\boldsymbol{\sigma})} B_{\tau(e)}, \qquad\qquad \nu^-(\tau) := \sum_{e \in \mathcal{E}^-(\boldsymbol{\sigma})} B_{\tau(e)}.$$

With these notations in place, and recalling the definitions of $\{p_{ij}\}_{i,j \in \{0,1\}}$ and $\{q_{ij}\}_{i,j \in \{0,1\}}$, the following lemma determines the posterior distribution of $\pi_*$ given $A$, $B$, and $\boldsymbol{\sigma}$.

**Lemma C.1** (Posterior distribution). *Let $\pi \in \mathcal{S}_n$ and let $\tau = \ell(\pi)$ be the corresponding lifted permutation. There is a constant $c = c(A, B, \boldsymbol{\sigma})$ such that*

$$\mathbb{P}\left(\pi_* = \pi \mid A, B, \boldsymbol{\sigma}\right) = c \left(\frac{p_{00}p_{11}}{p_{01}p_{10}}\right)^{\mu^+(\tau)_{11}} \left(\frac{q_{00}q_{11}}{q_{01}q_{10}}\right)^{\mu^-(\tau)_{11}} \left(\frac{p_{01}}{p_{00}}\right)^{\nu^+(\tau)} \left(\frac{q_{01}}{q_{00}}\right)^{\nu^-(\tau)}. \quad (50)$$

*Proof.* By Bayes' rule, we have that

$$\mathbb{P}\left(\pi_* = \pi \mid A, B, \boldsymbol{\sigma}\right) = \frac{\mathbb{P}\left(A, B \mid \pi_* = \pi, \boldsymbol{\sigma}\right) \mathbb{P}\left(\pi_* = \pi \mid \boldsymbol{\sigma}\right)}{\mathbb{P}\left(A, B \mid \boldsymbol{\sigma}\right)}.$$

Since the permutation $\pi_*$ is chosen uniformly at random and independently of the community labels, we have that $\mathbb{P}\left(\pi_* = \pi \mid \boldsymbol{\sigma}\right) = \mathbb{P}(\pi_* = \pi) = 1/n!$. Moreover, the term in the denominator only depends on $A$, $B$, and $\boldsymbol{\sigma}$ (it does not depend on $\pi$). We can therefore write

$$\mathbb{P}\left(\pi_* = \pi \mid A, B, \boldsymbol{\sigma}\right) = c(A, B, \boldsymbol{\sigma})\mathbb{P}\left(A, B \mid \pi_* = \pi, \boldsymbol{\sigma}\right),$$

where $c(A, B, \boldsymbol{\sigma}) = 1/\left(n! \, \mathbb{P}\left(A, B \mid \boldsymbol{\sigma}\right)\right)$. We now focus on computing $\mathbb{P}\left(A, B \mid \pi_* = \pi, \boldsymbol{\sigma}\right)$. Given $\boldsymbol{\sigma}$, the edge formation processes in the parent graph $G$ are mutually independent across

pairs of vertices. Since the subsampling procedure is also independent across pairs of vertices, we have that

$$\mathbb{P}(A, B \mid \pi_* = \pi, \boldsymbol{\sigma}) = \left( p_{00}^{\mu^+(\tau)_{00}} p_{01}^{\mu^+(\tau)_{01}} p_{10}^{\mu^+(\tau)_{10}} p_{11}^{\mu^+(\tau)_{11}} \right) \left( q_{00}^{\mu^-(\tau)_{00}} q_{01}^{\mu^-(\tau)_{01}} q_{10}^{\mu^-(\tau)_{10}} q_{11}^{\mu^-(\tau)_{11}} \right).$$
(51)

To simplify this expression, note that we can write

$$\mu^+(\tau)_{00} = \sum_{e \in \mathcal{E}^+(\boldsymbol{\sigma})} (1 - A_e) \left( 1 - B_{\tau(e)} \right) = \sum_{e \in \mathcal{E}^+(\boldsymbol{\sigma})} (1 - A_e) - \nu^+(\tau) + \mu^+(\tau)_{11},$$

$$\mu^+(\tau)_{01} = \sum_{e \in \mathcal{E}^+(\boldsymbol{\sigma})} (1 - A_e) B_{\tau(e)} = \nu^+(\tau) - \mu^+(\tau)_{11},$$

$$\mu^+(\tau)_{10} = \sum_{e \in \mathcal{E}^+(\boldsymbol{\sigma})} A_e \left( 1 - B_{\tau(e)} \right) = \sum_{e \in \mathcal{E}^+(\boldsymbol{\sigma})} A_e - \mu^+(\tau)_{11},$$

with similar expressions for $\mu^-(\tau)_{ij}$. The only terms on the right hand sides above that depend on $\tau$ (and therefore $\pi$) are $\nu^+(\tau)$ and $\mu^+(\tau)_{11}$; the remaining terms only depend on $A$ and $\boldsymbol{\sigma}$. We therefore have that

$$p_{00}^{\mu^+(\tau)_{00}} p_{01}^{\mu^+(\tau)_{01}} p_{10}^{\mu^+(\tau)_{10}} p_{11}^{\mu^+(\tau)_{11}} = C(A, \boldsymbol{\sigma}) \left( \frac{p_{00} p_{11}}{p_{01} p_{10}} \right)^{\mu^+(\tau)_{11}} \left( \frac{p_{01}}{p_{00}} \right)^{\nu^+(\tau)}$$

where $C(A, \boldsymbol{\sigma})$ depends only on $A$ and $\boldsymbol{\sigma}$. A similar expression holds for the other factor in (51), with $p_{ij}$ replaced with $q_{ij}$, $\mu^+(\tau)_{ij}$ replaced with $\mu^-(\tau)_{ij}$, and $\nu^+(\tau)$ replaced with $\nu^-(\tau)$. Plugging these back into (51) we obtain (50). □

Recall that $p_{00}, q_{00} = 1 - o(1)$ as $n \to \infty$, and that $p_{01}, p_{10}, p_{11}, q_{01}, q_{10}, q_{11}$ are all on the order $\log(n)/n$, implying that $p_{00} p_{11} > p_{01} p_{10}$ and $q_{00} q_{11} > q_{01} q_{10}$ for all $n$ large enough. Thus a useful consequence of Lemma C.1 is that $\mathbb{P}(\pi_* = \pi \mid A, B, \boldsymbol{\sigma})$ is increasing in $\mu^+(\tau)_{11}$ and $\mu^-(\tau)_{11}$, and decreasing in $\nu^+(\tau)$ and $\nu^-(\tau)$. Building on these observations, the following results establish conditions under which two lifted permutations, $\tau$ and $\tau'$, satisfy $\mu^+(\tau)_{11} \geq \mu^+(\tau')_{11}$ or $\nu^+(\tau) = \nu^+(\tau')$, with similar statements about $\mu^-$ and $\nu^-$. These will be used later to analyze the performance of the MAP estimator.

**Proposition C.2.** *Given* $\boldsymbol{\sigma}$, *the following holds. Let* $\pi_a, \pi_b \in \mathcal{S}_n$. *If*

$$\pi_a(V_+) = \pi_b(V_+) \qquad \text{and} \qquad \pi_a(V_-) = \pi_b(V_-),$$
(52)

*then* $\nu^+(\ell(\pi_a)) = \nu^+(\ell(\pi_b))$ *and* $\nu^-(\ell(\pi_a)) = \nu^-(\ell(\pi_b))$.

*Proof.* We prove the claim for $\nu^+$; the other claim follows from identical arguments. First note that $\mathcal{E}^+(\boldsymbol{\sigma}) = \binom{V_+}{2} \cup \binom{V_-}{2}$, so we can write

$$\nu^+(\ell(\pi_a)) = \sum_{(i,j) \in \binom{V_+}{2}} B_{\pi_a(i), \pi_a(j)} + \sum_{(i,j) \in \binom{V_-}{2}} B_{\pi_a(i), \pi_a(j)}.$$
(53)

In light of the assumption (52), the mapping $\pi_a^{-1} \circ \pi_b : V_+ \to V_+$ is a bijection. The first summation on the right hand side of (53) is therefore equal to

$$\sum_{(i,j) \in \binom{V_+}{2}} B_{\left( \pi_a \circ \pi_a^{-1} \circ \pi_b \right)(i), \left( \pi_a \circ \pi_a^{-1} \circ \pi_b \right)(j)} = \sum_{(i,j) \in \binom{V_+}{2}} B_{\pi_b(i), \pi_b(j)}.$$

Similarly, since $\pi_a^{-1} \circ \pi_b : V_- \to V_-$ is a bijection in light of (52), the second summation on the right hand side of (53) is equal to

$$\sum_{(i,j) \in \binom{V_-}{2}} B_{\pi_b(i), \pi_b(j)}.$$

Plugging the previous two displays back into (53) we obtain that $\nu^+(\ell(\pi_a)) = \nu^+(\ell(\pi_b))$. □

**Proposition C.3.** *Let* $\tau_a$ *and* $\tau_b$ *be lifted permutations such that whenever* $A_e B_{\tau_b(e)} = 1$ *we also have that* $A_e B_{\tau_a(e)} = 1$. *Then* $\mu^+(\tau_a)_{11} \geq \mu^+(\tau_b)_{11}$ *and* $\mu^-(\tau_a)_{11} \geq \mu^-(\tau_b)_{11}$.

*Proof.* The condition on $\tau_a$ and $\tau_b$ in the statement implies that $A_e B_{\tau_a(e)} \geq A_e B_{\tau_b(e)}$ for all $e \in \mathcal{E}$, and the desired result follows from the formulas for $\mu^+$ and $\mu^-$. □

## C.2 Performance of the MAP estimator and proof of Theorem 3.2

The following lemma shows how one may use the simple propositions above to bound the probability that the MAP estimator outputs a given permutation. Before stating the lemma, we recall a few properties of the MAP estimator. The estimator is formally given by

$$\widehat{\pi}_{\mathrm{MAP}} \in \arg\max_{\pi \in \mathcal{S}_n} \mathbb{P}\left(\pi_* = \pi \mid A, B, \boldsymbol{\sigma}\right). \tag{54}$$

In words, $\widehat{\pi}_{\mathrm{MAP}}$ is the mode of the posterior distribution $\{\mathbb{P}\left(\pi_* = \pi \mid A, B, \boldsymbol{\sigma}\right)\}_{\pi \in \mathcal{S}_n}$. When the argmax set is not a singleton, $\widehat{\pi}_{\mathrm{MAP}}$ is a uniform random element of the argmax set. The MAP estimator is *optimal*, in the sense that it minimizes the probability of error (see, e.g., [10, Chapter 4]).

For $\pi \in \mathcal{S}_n$, define the set

$$T^\pi \equiv T^\pi(A, B) := \left\{i \in [n] : \forall j \in [n], A_{i,j} B_{\pi(i),\pi(j)} = 0\right\},$$

as well as $T_+^\pi := T^\pi \cap V_+$ and $T_-^\pi := T^\pi \cap V_-$. Note that $T_+^\pi$ and $T_-^\pi$ are functions of $A$, $B$, and $\boldsymbol{\sigma}$. In words, if $\pi$ is the true vertex correspondence and $i \in T^\pi$, then the neighbors of $i$ in $G_1$ and the neighbors of $i$ in $G_2'$ are disjoint sets. Due to the lack of overlapping information, it becomes difficult for the MAP estimator to correctly match $i$ in $G_1$ with its counterpart $\pi(i)$ in $G_2$. The following lemma formalizes this (where we use the standard convention that $0! = 1$).

**Lemma C.4** (MAP estimator)**.** *For all $n$ large enough and for any $\pi \in \mathcal{S}_n$ we have that*

$$\mathbb{P}\left(\widehat{\pi}_{\mathrm{MAP}} = \pi \mid A, B, \boldsymbol{\sigma}\right) \leq \frac{1}{\left|T_+^\pi\right|! \cdot \left|T_-^\pi\right|!}.$$

*Proof.* Fix $\pi \in \mathcal{S}_n$ and suppose that $A$, $B$, and $\boldsymbol{\sigma}$ are given. Let $\rho_1$ be any permutation of $T_+^\pi$ and let $\rho_2$ be any permutation of $T_-^\pi$. Construct a new permutation $\pi' = \pi'\left(\pi, \rho_1, \rho_2\right)$ as follows:

- For $i \in [n] \setminus T^\pi$, let $\pi'(i) := \pi(i)$.

- For $i \in T_+^\pi$, let $\pi'(i) := \pi\left(\rho_1(i)\right)$.

- For $i \in T_-^\pi$, let $\pi'(i) := \pi\left(\rho_2(i)\right)$.

Let $\mathcal{T}$ be the set of permutations $\pi'$ constructed in this way. Since each choice of $\rho_1$ and $\rho_2$ leads to a distinct $\pi'$, we have that $|\mathcal{T}| = \left|T_+^\pi\right|! \cdot \left|T_-^\pi\right|!$.

A useful consequence of this construction is that $\pi'\left(V_+\right) = \pi\left(V_+\right)$ and $\pi'\left(V_-\right) = \pi\left(V_-\right)$. By Proposition C.2, this implies that

$$\nu^+\left(\ell\left(\pi'\right)\right) = \nu^+\left(\ell\left(\pi\right)\right) \qquad \text{and} \qquad \nu^-\left(\ell\left(\pi'\right)\right) = \nu^-\left(\ell\left(\pi\right)\right). \tag{55}$$

Furthermore, note that if $A_{i,j} B_{\pi(i),\pi(j)} = 1$, then we must have $i, j \in [n] \setminus T^\pi$ by definition. The construction of $\pi'$ implies that $\pi'(i) = \pi(i)$ and $\pi'(j) = \pi(j)$ for such $i$ and $j$. Hence we have that $A_{i,j} B_{\pi'(i),\pi'(j)} = A_{i,j} B_{\pi(i),\pi(j)} = 1$ for such $i$ and $j$. By Proposition C.3 we thus have that

$$\mu^+(\ell(\pi'))_{11} \geq \mu^+(\ell(\pi))_{11} \qquad \text{and} \qquad \mu^-(\ell(\pi'))_{11} \geq \mu^-(\ell(\pi))_{11}. \tag{56}$$

In light of Lemma C.1, as well as the observations on monotonicity made after its proof, (55) and (56) together imply, for all $n$ large enough, that

$$\mathbb{P}\left(\pi_* = \pi \mid A, B, \boldsymbol{\sigma}\right) \leq \mathbb{P}\left(\pi_* = \pi' \mid A, B, \boldsymbol{\sigma}\right). \tag{57}$$

Now we distinguish two cases. First, if $\pi$ is not a maximizer of $\{\mathbb{P}\left(\pi_* = \widetilde{\pi} \mid A, B, \boldsymbol{\sigma}\right)\}_{\widetilde{\pi} \in \mathcal{S}_n}$, then we have that $\mathbb{P}\left(\widehat{\pi}_{\mathrm{MAP}} = \pi \mid A, B, \boldsymbol{\sigma}\right) = 0$, so the claim holds trivially. On the other hand, if $\pi$ is a maximizer of $\{\mathbb{P}\left(\pi_* = \widetilde{\pi} \mid A, B, \boldsymbol{\sigma}\right)\}_{\widetilde{\pi} \in \mathcal{S}_n}$, then (by (57)) so is $\pi'$ for every $\pi' \in \mathcal{T}$. Therefore the set $\arg\max_{\widetilde{\pi} \in \mathcal{S}_n} \mathbb{P}\left(\pi_* = \widetilde{\pi} \mid A, B, \boldsymbol{\sigma}\right)$ has at least $|\mathcal{T}|$ elements. Since $\widehat{\pi}_{\mathrm{MAP}}$ picks an element of the argmax set uniformly at random, this implies that

$$\mathbb{P}\left(\widehat{\pi}_{\mathrm{MAP}} = \pi \mid A, B, \boldsymbol{\sigma}\right) \leq \frac{1}{|\mathcal{T}|} = \frac{1}{\left|T_+^\pi\right|! \cdot \left|T_-^\pi\right|!}. \qquad \square$$

Next, the following lemma establishes lower bounds for $\left|T_+^\pi\right|$ and $\left|T_-^\pi\right|$ in the case where $\pi$ is the ground truth vertex permutation. Before stating the result, for $\pi \in \mathcal{S}_n$ we define the measure $\mathbb{P}_\pi(\cdot) := \mathbb{P}(\cdot \mid \pi_* = \pi)$. Additionally, let $\mathbb{E}_\pi$ and $\operatorname{Var}_\pi$ denote the expectation and variance operators corresponding to the measure $\mathbb{P}_\pi$.

**Lemma C.5.** *Suppose that $s^2(\alpha + \beta)/2 < 1$. Then there exists $\gamma = \gamma(\alpha, \beta, s) > 0$ such that*

$$\lim_{n \to \infty} \min_{\pi \in \mathcal{S}_n} \mathbb{P}_\pi\left(\left|T_+^\pi\right|, \left|T_-^\pi\right| \geq n^\gamma\right) = 1.$$

The proof of the lemma is based on estimating the first and second moments of $\left|T_+^\pi\right|$ and $\left|T_-^\pi\right|$ under the measure $\mathbb{P}_\pi$. While the proof techniques are quite standard, the proof is somewhat tedious, so we defer it to Section C.3.

We are now ready to prove the impossibility result for graph matching in correlated SBMs.

*Proof of Theorem 3.2.* As mentioned before, we prove a stronger claim; namely, we show that even if $\boldsymbol{\sigma}$ is provided as extra information, for any estimator $\widetilde{\pi} = \widetilde{\pi}(G_1, G_2, \boldsymbol{\sigma})$ we have that $\lim_{n \to \infty} \mathbb{P}(\widetilde{\pi} = \pi_*) = 0$. To this end, we study the MAP estimator $\widehat{\pi}_{\mathrm{MAP}} = \widehat{\pi}_{\mathrm{MAP}}(A, B, \boldsymbol{\sigma})$ of $\pi_*$ given $A$, $B$, and $\boldsymbol{\sigma}$ (see (54)). Since the MAP estimator minimizes the probability of error (see, e.g., [10, Chapter 4]), it suffices to show that $\lim_{n \to \infty} \mathbb{P}(\widehat{\pi}_{\mathrm{MAP}} = \pi_*) = 0$.

To compute/bound $\mathbb{P}(\widehat{\pi}_{\mathrm{MAP}} = \pi_*)$, we may first condition on $\pi_*$ and then on $A$, $B$, and $\boldsymbol{\sigma}$. Since $\pi_* \in \mathcal{S}_n$ is uniformly random, we have that

$$\mathbb{P}(\widehat{\pi}_{\mathrm{MAP}} = \pi_*) = \frac{1}{n!} \sum_{\pi \in \mathcal{S}_n} \sum_{A, B, \boldsymbol{\sigma}} \mathbb{P}(\widehat{\pi}_{\mathrm{MAP}} = \pi \mid A, B, \boldsymbol{\sigma}, \pi_* = \pi) \mathbb{P}(A, B, \boldsymbol{\sigma} \mid \pi_* = \pi).$$

Note that $\widehat{\pi}_{\mathrm{MAP}}$ is a function of $A$, $B$, and $\boldsymbol{\sigma}$ (and perhaps additional randomness, in case the maximizer of the posterior distribution is not unique). Therefore $\mathbb{P}(\widehat{\pi}_{\mathrm{MAP}} = \pi \mid A, B, \boldsymbol{\sigma}, \pi_* = \pi) = \mathbb{P}(\widehat{\pi}_{\mathrm{MAP}} = \pi \mid A, B, \boldsymbol{\sigma})$, that is, we may remove the event $\{\pi_* = \pi\}$ from the conditioning. Plugging this back into the display above and using the bound of Lemma C.4 we obtain that

$$\mathbb{P}(\widehat{\pi}_{\mathrm{MAP}} = \pi_*) \leq \frac{1}{n!} \sum_{\pi \in \mathcal{S}_n} \mathbb{E}_\pi\left[\frac{1}{\left|T_+^\pi\right|! \cdot \left|T_-^\pi\right|!}\right], \tag{58}$$

where the expectation is over $A$, $B$, and $\boldsymbol{\sigma}$ (recall that $T_+^\pi$ and $T_-^\pi$ are functions of $A$, $B$, and $\boldsymbol{\sigma}$). Let $\gamma = \gamma(\alpha, \beta, s) > 0$ be the constant given by Lemma C.5, and for $\pi \in \mathcal{S}_n$ define the event $\mathcal{A}_\pi := \left\{\left|T_+^\pi\right|, \left|T_-^\pi\right| \geq n^\gamma\right\}$. By definition we have that

$$\mathbb{E}_\pi\left[\frac{1}{\left|T_+^\pi\right|! \cdot \left|T_-^\pi\right|!}\right] \leq \frac{1}{(n^\gamma!)^2} + \mathbb{P}_\pi(\mathcal{A}_\pi^c).$$

Plugging this into (58) we thus have that

$$\mathbb{P}(\widehat{\pi}_{\mathrm{MAP}} = \pi_*) \leq \frac{1}{(n^\gamma!)^2} + \frac{1}{n!} \sum_{\pi \in \mathcal{S}_n} \mathbb{P}_\pi(\mathcal{A}_\pi^c) \leq \frac{1}{(n^\gamma!)^2} + \max_{\pi \in \mathcal{S}_n} \mathbb{P}_\pi(\mathcal{A}_\pi^c).$$

Both terms on the right hand side go to 0 as $n \to \infty$; the latter term converging to 0 as $n \to \infty$ is due to Lemma C.5. $\qquad\square$

## C.3 Lower bounding $\left|T_+^\pi\right|$ and $\left|T_-^\pi\right|$: Proof of Lemma C.5

Fix $\pi \in \mathcal{S}_n$; throughout the proof we condition on the event $\{\pi_* = \pi\}$. Given also $\boldsymbol{\sigma}$, we have that

$$A_{i,j} B_{\pi(i), \pi(j)} \sim \begin{cases} \operatorname{Bernoulli}\left(s^2 \alpha \frac{\log n}{n}\right) & \text{if } (i,j) \in \mathcal{E}^+(\boldsymbol{\sigma}) \\ \operatorname{Bernoulli}\left(s^2 \beta \frac{\log n}{n}\right) & \text{if } (i,j) \in \mathcal{E}^-(\boldsymbol{\sigma}). \end{cases}$$

Moreover, for fixed $i \in [n]$ the random variables $\left\{A_{i,j} B_{\pi(i), \pi(j)}\right\}_{j \in [n] \setminus \{i\}}$ are mutually independent (given $\{\pi_* = \pi\}$ and $\boldsymbol{\sigma}$). Hence if $i \in V_+$, then we have that

$$\mathbb{P}_\pi(i \in T^\pi \mid \boldsymbol{\sigma}) = \left(1 - s^2 \alpha \frac{\log n}{n}\right)^{|V_+| - 1} \left(1 - s^2 \beta \frac{\log n}{n}\right)^{|V_-|}.$$

Note that $|V_+|$ and $|V_-|$ are typically approximately $n/2$, and hence the conditional probability above is typically approximately $n^{-s^2(\alpha+\beta)/2}$. To make this precise, we introduce some further notation. For $\epsilon \in (0,1)$ define

$$\delta := 1 - (1 + \epsilon/2)^2 s^2 (\alpha + \beta) / 2,$$
$$\lambda := 1 - (1 - \epsilon) s^2 (\alpha + \beta) / 2.$$

In the following we fix $\epsilon \in (0,1)$ such that

$$\delta > 0 \qquad \text{and} \qquad \lambda > 0 \qquad \text{and} \qquad \lambda < 2\delta. \tag{59}$$

Such an $\epsilon \in (0,1)$ exists due to the assumption that $s^2 (\alpha + \beta) / 2 < 1$. Recall that on the event $\mathcal{F}_\epsilon$ we have that $|V_+|, |V_-| \leq (1 + \epsilon/2)n/2$. Thus if $\boldsymbol{\sigma}$ is such that the event $\mathcal{F}_\epsilon$ holds, then

$$\log \mathbb{P}_\pi (i \in T^\pi \,|\, \boldsymbol{\sigma}) \geq \left(1 + \frac{\epsilon}{2}\right) \frac{n}{2} \left( \log \left(1 - s^2 \alpha \frac{\log n}{n}\right) + \log \left(1 - s^2 \beta \frac{\log n}{n}\right) \right)$$

$$\geq \left(1 + \frac{\epsilon}{2}\right)^2 \frac{n}{2} \left( -s^2(\alpha + \beta) \frac{\log n}{n} \right) = (\delta - 1) \log n,$$

where the second inequality holds for all $n$ large enough, since $\log(1 - x) \geq -(1 + \epsilon/2)x$ for all $x > 0$ small enough. Thus, on the event $\mathcal{F}_\epsilon$ we have that $\mathbb{P}_\pi (i \in T^\pi \,|\, \boldsymbol{\sigma}) \geq n^{\delta-1}$ for all $n$ large enough. By linearity of expectation this gives a lower bound on the (conditional) expectation of $|T_+^\pi|$: if $\boldsymbol{\sigma}$ is such that $\mathcal{F}_\epsilon$ holds, then for all $n$ large enough we have that

$$\mathbb{E}_\pi \left[ |T_+^\pi| \,\big|\, \boldsymbol{\sigma} \right] \geq |V_+| n^{\delta-1} \geq \frac{1 - \epsilon/2}{2} n^\delta \geq \frac{1}{4} n^\delta. \tag{60}$$

To establish a probabilistic lower bound for $|T_+^\pi|$, we proceed by bounding its (conditional) variance. For $i \in [n]$ let $X_i := \mathbf{1}\left(i \in T_+^\pi\right)$ be the indicator variable that $i \in T_+^\pi$. We then have that

$$\text{Var}_\pi \left( |T_+^\pi| \,\big|\, \boldsymbol{\sigma} \right) = \text{Var}_\pi \left( \sum_{i \in V_+} X_i \,\bigg|\, \boldsymbol{\sigma} \right) = \sum_{i \in V_+} \text{Var}_\pi \left( X_i \,|\, \boldsymbol{\sigma} \right) + \sum_{i,j \in V_+ : i \neq j} \text{Cov}_\pi \left( X_i, X_j \,|\, \boldsymbol{\sigma} \right). \tag{61}$$

For the variance terms on the right hand side, we use the bound

$$\text{Var}_\pi \left( X_i \,|\, \boldsymbol{\sigma} \right) \leq \mathbb{P}_\pi \left( i \in T^\pi \,|\, \boldsymbol{\sigma} \right) \leq \exp \left( -s^2 \left( \alpha \left( |V_+| - 1 \right) + \beta \left| V_- \right| \right) \frac{\log n}{n} \right).$$

If $\boldsymbol{\sigma}$ is such that $\mathcal{F}_\epsilon$ holds, then using the bounds $|V_+| - 1 \geq (1 - \epsilon)n/2$ and $|V_-| \geq (1 - \epsilon)n/2$ we thus have that

$$\text{Var}_\pi \left( X_i \,|\, \boldsymbol{\sigma} \right) \leq n^{\lambda-1}. \tag{62}$$

The covariance terms can be computed as

$$\text{Cov}_\pi \left( X_i, X_j \,|\, \boldsymbol{\sigma} \right) = \mathbb{E}_\pi \left[ X_i X_j \,|\, \boldsymbol{\sigma} \right] - \mathbb{E}_\pi \left[ X_i \,|\, \boldsymbol{\sigma} \right] \mathbb{E}_\pi \left[ X_j \,|\, \boldsymbol{\sigma} \right]$$

$$= \left(1 - s^2 \alpha \frac{\log n}{n}\right)^{2|V_+|-3} \left(1 - s^2 \beta \frac{\log n}{n}\right)^{2|V_-|} - \left(1 - s^2 \alpha \frac{\log n}{n}\right)^{2|V_+|-2} \left(1 - s^2 \beta \frac{\log n}{n}\right)^{2|V_-|}$$

$$= s^2 \alpha \frac{\log n}{n} \left(1 - s^2 \alpha \frac{\log n}{n}\right)^{2|V_+|-3} \left(1 - s^2 \beta \frac{\log n}{n}\right)^{2|V_-|}$$

$$\leq s^2 \alpha \frac{\log n}{n} \exp \left( -s^2 \left( \alpha \left( 2 |V_+| - 3 \right) + \beta \left( 2 |V_-| \right) \right) \frac{\log n}{n} \right).$$

If $\boldsymbol{\sigma}$ is such that $\mathcal{F}_\epsilon$ holds, then using the bounds $2 |V_+| - 3 \geq (1 - \epsilon)n$ and $2 |V_-| \geq (1 - \epsilon)n$ we thus have that

$$\text{Cov}_\pi \left( X_i, X_j \,|\, \boldsymbol{\sigma} \right) \leq \left( s^2 \alpha \log(n) \right) n^{-1-(1-\epsilon)s^2(\alpha+\beta)} = \left( s^2 \alpha \log(n) \right) n^{2\lambda-3}. \tag{63}$$

Plugging (62) and (63) back into (61), we have that

$$\text{Var}_\pi \left( |T_+^\pi| \,\big|\, \boldsymbol{\sigma} \right) \leq n \cdot n^{\lambda-1} + n^2 \cdot \left( s^2 \alpha \log(n) \right) n^{2\lambda-3} = n^\lambda + \left( s^2 \alpha \log(n) \right) n^{2\lambda-1}.$$

whenever $\boldsymbol{\sigma}$ is such that $\mathcal{F}_\epsilon$ holds. Since $\lambda < 1$, we have that $\lambda > 2\lambda - 1$, and so the display above implies that

$$\mathrm{Var}_\pi\left(\left|T_+^\pi\right| \,\middle|\, \boldsymbol{\sigma}\right) \leq 2n^\lambda \tag{64}$$

for all $n$ large enough, whenever $\boldsymbol{\sigma}$ is such that $\mathcal{F}_\epsilon$ holds.

Next, we use Chebyshev's inequality to turn the first and second moment estimates into a probabilistic lower bound for $\left|T_+^\pi\right|$. If $\boldsymbol{\sigma}$ is such that $\mathcal{F}_\epsilon$ holds, then, by (60), for all $n$ large enough such that $n^{\delta/2} \leq n^\delta/8$, we have that

$$\mathbb{P}_\pi\left(\left|T_+^\pi\right| \leq n^{\delta/2} \,\middle|\, \boldsymbol{\sigma}\right) \leq \mathbb{P}_\pi\left(\left|\left|T_+^\pi\right| - \mathbb{E}_\pi\left[\left|T_+^\pi\right| \,\middle|\, \boldsymbol{\sigma}\right]\right| \geq n^\delta/8 \,\middle|\, \boldsymbol{\sigma}\right).$$

Thus by Chebyshev's inequality and (64) we have that

$$\mathbb{P}_\pi\left(\left|T_+^\pi\right| \leq n^{\delta/2} \,\middle|\, \boldsymbol{\sigma}\right) \leq 64n^{-2\delta}\,\mathrm{Var}_\pi\left(\left|T_+^\pi\right| \,\middle|\, \boldsymbol{\sigma}\right) \leq 128n^{\lambda-2\delta}$$

for all $n$ large enough, whenever $\boldsymbol{\sigma}$ is such that $\mathcal{F}_\epsilon$ holds. Recall from (59) that $\lambda - 2\delta < 0$, so this bound decays to 0 as $n \to \infty$.

To remove the conditioning on $\boldsymbol{\sigma}$, we can write

$$\mathbb{P}_\pi\left(\left|T_+^\pi\right| \geq n^{\delta/2}\right) \geq \mathbb{E}\left[\mathbb{P}_\pi\left(\left|T_+^\pi\right| \geq n^{\delta/2} \,\middle|\, \boldsymbol{\sigma}\right)\mathbf{1}\left(\mathcal{F}_\epsilon\right)\right] \geq \left(1 - 128n^{\lambda-2\delta}\right)\mathbb{P}\left(\mathcal{F}_\epsilon\right).$$

Note in particular that this lower bound holds *uniformly* over all $\pi \in \mathcal{S}_n$. Hence, since $\mathbb{P}\left(\mathcal{F}_\epsilon\right) \to 1$ as $n \to \infty$, we have that

$$\lim_{n\to\infty} \min_{\pi\in\mathcal{S}_n} \mathbb{P}_\pi\left(\left|T_+^\pi\right| \geq n^{\delta/2}\right) = 1.$$

Finally, the same arguments also hold for $\left|T_-^\pi\right|$ by symmetry, so the conclusion follows by a union bound.

## D   Impossibility of community recovery from correlated SBMs

*Proof of Theorem 3.4.* The key idea is to reduce the problem to that of exact community recovery in the (classical) single-graph SBM setting. Specifically, as observed in the proof of Theorem 3.3 in the main text, the union graph $H_* := G_1 \vee_{\pi_*} G_2$ satisfies

$$H_* \sim \mathrm{SBM}\left(n, \alpha(1-(1-s)^2)\frac{\log n}{n}, \beta(1-(1-s)^2)\frac{\log n}{n}\right),$$

and from $H_*$ it is possible to simulate $G_1$ and $G_2$. However, under the condition

$$\left|\sqrt{\alpha} - \sqrt{\beta}\right| < \sqrt{\frac{2}{1-(1-s)^2}}, \tag{65}$$

exact community recovery is impossible from an SBM with such parameters [2, 8, 3, 1].

To make the argument formal, suppose by way of contradiction that there exists an estimator $\widetilde{\boldsymbol{\sigma}} = \widetilde{\boldsymbol{\sigma}}(G_1, G_2)$ such that

$$\limsup_{n\to\infty} \mathbb{P}\left(\mathrm{ov}\left(\widetilde{\boldsymbol{\sigma}}(G_1, G_2), \boldsymbol{\sigma}\right) = 1\right) > 0. \tag{66}$$

Now let $H$ be a graph on the vertex set $[n]$ satisfying

$$H \sim \mathrm{SBM}\left(n, \alpha(1-(1-s)^2)\frac{\log n}{n}, \beta(1-(1-s)^2)\frac{\log n}{n}\right),$$

and let $\boldsymbol{\sigma}_H$ denote the underlying community labels of $H$. Given $H$, we now construct two edge-subsampled graphs $H_1$ and $H_2'$ as follows. First, define the parameters

$$(r_{01}, r_{10}, r_{11}) := \left(\frac{s(1-s)}{1-(1-s)^2}, \frac{s(1-s)}{1-(1-s)^2}, \frac{s^2}{1-(1-s)^2}\right)$$

and note that $r_{01} + r_{10} + r_{11} = 1$, so this triple defines a probability distribution. Now for every vertex pair $(i, j)$ independently:

- if $(i, j)$ is not an edge in $H$, then it is not an edge in $H_1$ and it is not an edge in $H_2'$;

- if $(i, j)$ is an edge in $H$, then

  – with probability $r_{10}$, the pair $(i, j)$ is an edge in $H_1$ but not an edge in $H_2'$;
  – with probability $r_{01}$, the pair $(i, j)$ is not an edge in $H_1$ but it is an edge in $H_2'$; and
  – with probability $r_{11}$, the pair $(i, j)$ is an edge in both $H_1$ and $H_2'$.

The key observation is that, by construction, $(H_1, H_2', \boldsymbol{\sigma}_H)$ has the same distribution as $(G_1, G_2', \boldsymbol{\sigma})$. Now let $\pi \in \mathcal{S}_n$ be a uniformly random permutation which is independent of everything else. Finally, we generate $H_2$ by relabeling the vertices of $H_2'$ according to $\pi$ (i.e., vertex $i$ in $H_2'$ is relabeled to $\pi(i)$ in $H_2$). Again by construction, $(H_1, H_2, \boldsymbol{\sigma}_H)$ has the same distribution as $(G_1, G_2, \boldsymbol{\sigma})$. In particular, ov $(\widetilde{\boldsymbol{\sigma}}(H_1, H_2), \boldsymbol{\sigma}_H)$ and ov $(\widetilde{\boldsymbol{\sigma}}(G_1, G_2), \boldsymbol{\sigma})$ have the same distribution, and so

$$\mathbb{P}\left(\text{ov}\left(\widetilde{\boldsymbol{\sigma}}(H_1, H_2), \boldsymbol{\sigma}_H\right) = 1\right) = \mathbb{P}\left(\text{ov}\left(\widetilde{\boldsymbol{\sigma}}(G_1, G_2), \boldsymbol{\sigma}\right) = 1\right).$$

Combining this with (66), we have that

$$\limsup_{n \to \infty} \mathbb{P}\left(\text{ov}\left(\widetilde{\boldsymbol{\sigma}}(H_1, H_2), \boldsymbol{\sigma}_H\right) = 1\right) > 0. \tag{67}$$

However, it is known [2, 8, 3, 1] that if (65) holds, then for every estimator $\boldsymbol{\sigma}' = \boldsymbol{\sigma}'(H)$ (including randomized estimators) we have that

$$\lim_{n \to \infty} \mathbb{P}\left(\text{ov}\left(\boldsymbol{\sigma}'(H), \boldsymbol{\sigma}_H\right) = 1\right) = 0. \tag{68}$$

Since $(H_1, H_2)$ was constructed from $H$ using only additional randomness, $\widetilde{\boldsymbol{\sigma}}(H_1, H_2)$ can be thought of as a randomized estimator of $\boldsymbol{\sigma}_H$ which takes $H$ as input. Therefore (67) and (68) are in direct contradiction. Thus (66) does not hold, which proves the claim. $\qquad\square$

## E  Proofs for many correlated SBMs

In this section we prove our results that concern $K \geq 3$ correlated SBMs, namely Theorems 3.6 and 3.7. These proofs are analogous to the proofs of Theorems 3.3 and 3.4, extending them to the setting of $K \geq 3$ correlated SBMs.

*Proof of Theorem 3.6.* Given permutations $\pi^2, \ldots, \pi^K \in \mathcal{S}_n$, we define $G_1 \vee_{\pi^2} G_2 \ldots \vee_{\pi^K} G_K$, the *union graph with respect to* $\pi^2, \ldots, \pi^K$, as follows: for distinct $i$ and $j$, the pair $(i, j)$ is an edge in $G_1 \vee_{\pi^2} G_2 \ldots \vee_{\pi^K} G_K$ if and only if $(i, j)$ is an edge in $G_1$ or $\left(\pi^k(i), \pi^k(j)\right)$ is an edge in $G_k$ for some $k \in \{2, \ldots, K\}$. In particular, let $H_* := G_1 \vee_{\pi_*^2} G_2 \ldots \vee_{\pi_*^K} G_K$. By construction, $H_*$ is the subgraph of the parent graph $G$ consisting of exactly the edges that are in $G_1$ or in $G_k'$ for some $k \in \{2, \ldots, K\}$. Thus we have that

$$H_* \sim \text{SBM}\left(n, \alpha\left(1 - (1-s)^K\right)\frac{\log n}{n}, \beta\left(1 - (1-s)^K\right)\frac{\log n}{n}\right).$$

The algorithm we study first computes, for every $k \in \{2, \ldots, K\}$, the permutation $\widehat{\pi}^k := \widehat{\pi}\left(G_1, G_k\right)$ according to Theorem 3.1. We then pick any community recovery algorithm that is known to succeed until the information-theoretic limit, and run it on $\widehat{H} := G_1 \vee_{\widehat{\pi}^2} G_2 \ldots \vee_{\widehat{\pi}^K} G_K$; we denote the result of this algorithm by $\widehat{\sigma}(\widehat{H})$. We can then write

$$\mathbb{P}(\text{ov}(\widehat{\boldsymbol{\sigma}}(\widehat{H}), \boldsymbol{\sigma}) \neq 1) \leq \mathbb{P}(\{\text{ov}(\widehat{\boldsymbol{\sigma}}(\widehat{H}), \boldsymbol{\sigma}) \neq 1\} \cap \{\widehat{H} = H_*\}) + \mathbb{P}(\widehat{H} \neq H_*)$$

$$\leq \mathbb{P}(\text{ov}(\widehat{\boldsymbol{\sigma}}(H_*), \boldsymbol{\sigma}) \neq 1) + \sum_{k=2}^{K} \mathbb{P}\left(\widehat{\pi}^k \neq \pi_*^k\right),$$

where, to obtain the inequality in the second line, we have used that $\widehat{\boldsymbol{\sigma}}(\widehat{H}) = \widehat{\boldsymbol{\sigma}}(H_*)$ on the event $\{\widehat{H} = H_*\}$, and that $\widehat{H} \neq H_*$ implies that $\widehat{\pi}^k \neq \pi_*^k$ for some $k \in \{2, \ldots, K\}$. Since exact community recovery on $H_*$ is possible when the condition

$$\left|\sqrt{\alpha} - \sqrt{\beta}\right| > \sqrt{\frac{2}{1 - (1-s)^K}}$$

holds [2, 8, 3, 1], we know that $\mathbb{P}(\text{ov}(\widehat{\boldsymbol{\sigma}}(H_*), \boldsymbol{\sigma}) \neq 1) \to 0$ as $n \to \infty$. In light of Theorem 3.1 we also have, for every $k \in \{2, \dots, K\}$, that $\mathbb{P}(\widehat{\pi}^k \neq \pi_*^k) \to 0$ when $s^2(\alpha + \beta)/2 > 1$, concluding the proof. $\qquad\square$

*Proof of Theorem 3.7.* Suppose, by way of contradiction, that there exists an estimator $\widetilde{\boldsymbol{\sigma}} = \widetilde{\boldsymbol{\sigma}}(G_1, G_2, \dots, G_K)$ such that

$$\limsup_{n \to \infty} \mathbb{P}\left(\text{ov}\left(\widetilde{\boldsymbol{\sigma}}(G_1, G_2, \dots, G_K), \boldsymbol{\sigma}\right) = 1\right) > 0. \tag{69}$$

Now let $H$ be a graph on the vertex set $[n]$ satisfying

$$H \sim \text{SBM}\left(n, \alpha\left(1 - (1-s)^K\right)\frac{\log n}{n}, \beta\left(1 - (1-s)^K\right)\frac{\log n}{n}\right),$$

and let $\boldsymbol{\sigma}_H$ denote the underlying community labels of $H$. Given $H$, we now construct $K$ edge-subsampled graphs, $H_1, H_2', \dots, H_K'$, as follows. First, for $x \in \{0,1\}^K$ let $|x| := \sum_{k=1}^K x_k$. For every $x \in \{0,1\}^K$ let $r_x := s^{|x|}(1-s)^{K-|x|}/\left(1 - (1-s)^K\right)$, and note that $\sum_{x \in \{0,1\}^K \setminus 0^K} r_x = 1$, so $\boldsymbol{r} := \{r_x\}_{x \in \{0,1\}^K \setminus 0^K}$ defines a probability distribution. Now for every vertex pair $(i, j)$ independently:

- if $(i, j)$ is not an edge in $H$, then it is not an edge in any of $H_1, H_2', \dots, H_K'$;

- if $(i, j)$ is an edge in $H$, then draw $x \in \{0,1\}^K \setminus 0^K$ from the distribution $\boldsymbol{r}$. Then $(i, j)$ is an edge in $H_1$ if and only if $x_1 = 1$, and for every $k \in \{2, \dots, K\}$, the pair $(i, j)$ is an edge in $H_k'$ if and only if $x_k = 1$.

The key observation is that, by construction, $(H_1, H_2', \dots, H_K', \boldsymbol{\sigma}_H)$ has the same distribution as $(G_1, G_2', \dots, G_K', \boldsymbol{\sigma})$. Now let $\pi^2, \dots, \pi^K \in \mathcal{S}_n$ be i.i.d. uniformly random permutations which are independent of everything else. Finally, for every $k \in \{2, \dots, K\}$, we generate $H_k$ by relabeling the vertices of $H_k'$ according to $\pi^k$ (i.e., vertex $i$ in $H_k'$ is relabeled to $\pi^k(i)$ in $H_k$). Again by construction, $(H_1, H_2, \dots, H_K, \boldsymbol{\sigma}_H)$ has the same distribution as $(G_1, G_2, \dots, G_K, \boldsymbol{\sigma})$. In particular, $\text{ov}\left(\widetilde{\boldsymbol{\sigma}}(H_1, H_2, \dots, H_K), \boldsymbol{\sigma}_H\right)$ and $\text{ov}\left(\widetilde{\boldsymbol{\sigma}}(G_1, G_2, \dots, G_K), \boldsymbol{\sigma}\right)$ have the same distribution, and so

$$\mathbb{P}\left(\text{ov}\left(\widetilde{\boldsymbol{\sigma}}(H_1, H_2, \dots, H_K), \boldsymbol{\sigma}_H\right) = 1\right) = \mathbb{P}\left(\text{ov}\left(\widetilde{\boldsymbol{\sigma}}(G_1, G_2, \dots, G_K), \boldsymbol{\sigma}\right) = 1\right).$$

Combining this with (69), we have that

$$\limsup_{n \to \infty} \mathbb{P}\left(\text{ov}\left(\widetilde{\boldsymbol{\sigma}}(H_1, H_2, \dots, H_K), \boldsymbol{\sigma}_H\right) = 1\right) > 0. \tag{70}$$

However, it is known [2, 8, 3, 1] that if the condition

$$\left|\sqrt{\alpha} - \sqrt{\beta}\right| < \sqrt{\frac{2}{1 - (1-s)^K}}$$

holds, then for every estimator $\boldsymbol{\sigma}' = \boldsymbol{\sigma}'(H)$ (including randomized estimators) we have that

$$\lim_{n \to \infty} \mathbb{P}\left(\text{ov}\left(\boldsymbol{\sigma}'(H), \boldsymbol{\sigma}_H\right) = 1\right) = 0. \tag{71}$$

Since $(H_1, H_2, \dots, H_K)$ was constructed from $H$ using only additional randomness, the estimator $\widetilde{\boldsymbol{\sigma}}(H_1, H_2, \dots, H_K)$ is a randomized estimator of $\boldsymbol{\sigma}_H$ which takes $H$ as input. Therefore (70) and (71) are in direct contradiction. Thus (69) does not hold, proving the claim. $\qquad\square$