# OpenReview forum: "Correlated Stochastic Block Models: Exact Graph Matching with Applications to Recovering Communities"
_NeurIPS.cc/2021/Conference — NeurIPS 2021 Spotlight_

### Official Review · Reviewer_maHr · 2021-07-09

**Rating:** 7
**Confidence:** 4

**Summary:**

This paper provides necessary and sufficient conditions for exact recovery problems in a two-community correlated stochastic block model (SBMs). In this model, correlated SBMs are generated on the same set of vertices in a natural way and then the vertices of  one of the SBMs is randomly permuted. The paper gives necessary and sufficient conditions for 1) recovering the (unknown) permutation to recover the correspondence between the two SBMs and 2) recovering the underlying community memberships.  The extension to multiple correlated SBMs in also considered.

**Limitations And Societal Impact:**

The assumptions for the theoretical results are clearly stated.  Given the theoretical nature of this work I do not see the need for any more discussion on societal impact.

**Main Review:**

This paper is clearly written. It provides a nice descriptions of the problem and the results are easy to follow. Overall, I enjoyed reading it.

The key part of the proof requires a careful analysis of the conditions for finding the graph matching. While it seems the main approach builds upon previous work, there are a number of technical details which are well explained in the supplementary material. I think this analysis should also be viewed as a useful contribution of the paper.

The significance and novelty are on par with related work in the literature. Also, similar to other work in the literature, the model is not very realistic so these results are more an exercise in mathematical analysis rather than shedding new insight into community detection problems in practice. Along these lines, one may question whether exact community recovery is a realistic objective, as opposed to approximate recovery. The authors do mention approximate recovery in the future work section.


**Time Spent Reviewing:**

3

---

> ### Author Response · Authors · 2021-08-10
> **Approximate recovery**
>
> Thank you for your review! The question of approximate community recovery is an interesting one. A related question of possible interest to you is whether approximate recovery of the vertex correspondence is possible. Our results imply that in the logarithmic degree regime we consider, it is always possible to correctly match a (1 - o(1)) fraction of vertices - see Lemma C.9 in the Supplementary Material. (Since our focus is on exact graph matching and community recovery, we did not highlight this as a main result, however.)
>
> Coming back to the approximate community recovery problem, we conjecture that one can use an approximate matching (one that correctly matches a (1-o(1)) fraction of vertices) instead of an exact matching to recovery communities either approximately or exactly -- see Conjecture 3.5 and the discussion following it for details. We remark that it is unclear how one can leverage an approximate matching to obtain exact or approximate community recovery -- we shall investigate this in future work.

---

### Official Review · Reviewer_dpAb · 2021-07-16

**Rating:** 7
**Confidence:** 3

**Summary:**

This paper studies the exact graph matching problem in correlated stochastic block model. With the exact graph matching function, classical community detection algorithm can achieve exact community recovery.

**Limitations And Societal Impact:**

The author has clearly indicated the limitation of this paper. The contribution to the community recovery problem is actually limited. The paper has not solved the community detection problem. Given the true matching function of the nodes, community recovery can be obtained by existing algorithms.

**Main Review:**

The main contribution of the this paper is the analysis on phase transaction on the possibility on exact graph matching in correlated stochastic block model. Existing literature only solves the problem with correct community labels. The result is interesting and substantial in network analysis.
The paper is well organized and reader-friendly.

**Time Spent Reviewing:**

2

---

> ### Author Response · Authors · 2021-08-10
> **Information-theoretic thresholds for exact community recovery**
>
> Thank you for your review! While your assessment of the limitations of our work is correct, we would like to add a few related remarks. For certain parameter regimes, our converse and achievability results actually yield matching bounds -- see for instance Figure 2a in the Supplementary Material.
>
> You are also correct in stating "given the true matching function of the nodes, community recovery can be obtained by existing algorithms". In Conjecture 3.5 on page 6, we conjecture that exact community recovery is possible even when exact graph matching is impossible. To handle this case, we may need to design new algorithms for community recovery using a partial (rather than exact) matching between vertices of G1 and G2. For more details, see the discussion after Conjecture 3.5 and in lines 323-329, page 9. We leave this as an objective for future work, however.

---

### Official Review · Reviewer_DFob · 2021-07-19

**Rating:** 8
**Confidence:** 5

**Summary:**

This paper revisits the community detection problem from a theoretical perspective. The community detection problem on the SBM is now well-understood, this paper looks at a case where 2 correlated SBMs are given instead of only one. The nodes of the 2 SBMs represent the same population but the mapping between the nodes of the 2 SBMs is not known. This is a natural extension of the mathematical problem of community detection on a single SBM motivated for example by de-anonymizing social networks.
The authors obtain partial results for exact recovery. A natural algorithm is to first align the 2 SBMs and then used the union of the 2 SBMs which is still a SBM to find the communities. The authors fully characterize when exact alignment is possible and then provide sufficient conditions for exact recovery. They also prove a partial converse result and provide conditions under which community recovery is information theoretically impossible.

**Limitations And Societal Impact:**

OK

**Main Review:**

The paper is very clear, well written and easy to read. I think that this line of work is a very natural extension for the SBM and will lead to subsequent works. As the authors claim, their work is a first step and I suspect that it will trigger interest in the crowd working on the theory of ML.

**Time Spent Reviewing:**

3

---

> ### Author Response · Authors · 2021-08-10
> **Thanks**
>
> Thank you for your review!

---

### Official Review · Reviewer_aMdS · 2021-07-19

**Rating:** 8
**Confidence:** 3

**Summary:**

This paper studies the task of learning latent community structure from two correlated Stochastic Block Models (SBM). They provide a information theoretic threshold above which there exists a estimator that learns the community structure with probability 1, and below which no estimator can recover the true correspondence with some probability away from 0.

**Limitations And Societal Impact:**

Yes

**Main Review:**

This is a well written paper that not only is a precursor for using multiple correlated graphs to learn underlying latent communities, it demonstrates the advantages of using multiple graphs instead of a single graph (Theorem 3.6). The authors have shown robust and novel understanding of the subject as well as described conjectures on where future research work in this area could potentially head. I find that this adds to the significance of their results. Clear accept.

**Time Spent Reviewing:**

7

---

> ### Author Response · Authors · 2021-08-10
> **Thanks**
>
> Thank you for your review!

---

### Decision · Program_Chairs · 2021-09-27

**Decision:**

Accept (Spotlight)

**Comment:**

This paper presents a fine theoretical contribution showing how an initial step of graph matching of correlated graphs yields improved results to subsequently perform community detection, assuming graphs sampled from a correlated stochastic block model. The reviews are quite positive. Indeed joint consideration of multiple graphs to perform tasks such as community detection is a very natural strategy, and does not seem to have received much attention in the past. This makes the paper's contribution all the more valuable, hence the recommendation to accept.